# East Asian summer rainfall stimulated by subseasonal Indian monsoonal heating

Shixue Li [1] ✉, Tomonori Sato [2], Tetsu Nakamura [2,4] & Wenkai Guo[3]

The responses of the East Asian summer monsoon (EASM) to the Indian summer monsoon (ISM) have been the subject of extensive investigation. Nevertheless, it remains uncertain whether the ISM can serve as a predictor for the EASM. Here, on the basis of both observations and a large-ensemble climate model experiment, we show that the subseasonal variability of abnormal diabatic heating over India enhances precipitation over central East China, the Korean Peninsula, and southern Japan in June. ISM heating triggers Rossby wave propagation along the subtropical jet, promoting southerly winds over East Asia. The southerly winds helps steer anomalous mid-tropospheric warm advection and lower-tropospheric moisture advection toward East Asia, providing conditions preferential for rainband formation. Cluster analysis shows that, depending on jet structures, ISM heating can serve as a trigger as well as a reinforcer of the rainband.

The East Asian summer monsoon (EASM) is one of the most prominent components of the Asian summer monsoon system. The EASM is characterized as a northeastward extension of a rainband that results in spatiotemporally continuous rainfall, affecting East China, the Korean Peninsula, and Japan in summer[1,2]. The rainband migrates northward from early summer to late summer (i.e., June–August) with the largest areal coverage and greatest amount of banded precipitation occurring in June[3]. As a subtropical monsoon, the EASM is influenced by multiple remote factors that originate from tropical[4–7], mid-latitude[8–10], and even polar regions[11–13]. Thus, making predictions regarding the EASM rainband remains challenging[14–16] despite the increasing levels of attention focused on related extreme weather events that include droughts and floods[17,18]. Consequently, unanticipated EASM events could have devastating influence on agriculture, water resources, and human health in East Asian countries[19] if appropriate mitigation strategies are not undertaken based on operational forecasts.

Another principal component of the Asian summer monsoon system is the Indian summer monsoon (ISM), which is independent of the EASM despite their recognized interaction[20–22]. The rainfall and resultant diabatic heating of the ISM trigger a Rossby wave train[23], known as circumglobal teleconnection (CGT) that is manifested as a leading mode of the summer circulation over mid-latitude Eurasia with zonal wavenumber-5 structure[22,24,25]. In CGT episodes, the 200-hPa geopotential height over East Asia, the North Pacific, North America, and the Northeast Atlantic are all nearly in phase and synchronized with the geopotential height variations over the ISM area, specifically over the northern Indian subcontinent and West-to-Central Asia. Through such covariation, the CGT affects the weather[26] and climate[22,27] across the Northern Hemisphere. Hence, it is reasonable to hypothesize that the CGT acts as a bridge in terms of precipitation variability between the ISM and the EASM. Improved understanding of the ISM–EASM bridge could add value to EASM predictions. However, summer precipitation in these two monsoon regions does not have long-term correlation on the interannual timescale; their correlation appears only in certain periods or in different subregions other than in relation to the rainband in East Asia[28–30]. This unstable relationship is perhaps attributable to variations of the subtropical westerly jet, which acts as a waveguide of the CGT[29], and weakening of the ISM–El Niño relationship[31], which has diminished the ISM and its associated CGT variability since the mid-1970s[32]. Hence, there is currently no agreement on whether the ISM can be regarded as a robust predictor for the EASM. Considering the complex influencing factors of the Asian summer monsoon systems that operate across multiple time scales,

[1]Graduate School of Environmental Science, Hokkaido University, Sapporo 060-0810, Japan. [2]Faculty of Environmental Earth Science, Hokkaido University, Sapporo 060-0810, Japan. [3]Faculty of Geosciences and Environmental Engineering, Southwest Jiaotong University, Chengdu 611756, China. [4]Present address: Climate Prediction Division, Japan Meteorological Agency, Tokyo 105-8431, Japan. ✉e-mail: lishx@ees.hokudai.ac.jp

treatments of monthly and seasonal averaging might have the effect of obscuring the dynamical relationship between the ISM and the EASM. Therefore, we conducted analysis at the subseasonal timescale, which is more appropriate for determining physical linkages between the ISM and the EASM.

Here, we reveal that the EASM rainband in June is modulated by diabatic heating within the ISM region on the subseasonal timescale. We further classify the subtropical jet patterns to discuss how the ISM's heating signal propagates to the EASM domain.

## Results

### Subseasonal synchronization between ISM and EASM

In this section, the ISM–EASM causal connection is examined through lead–lag composite analysis with respect to extreme Indian monsoonal heating (lag0 day; see "Methods") on the subseasonal timescale, based on an index representing the ISM heating released to the troposphere (hereafter ISMH, see "Methods"). The observations and a large-ensemble historical climate experiment (hereafter HIST, see "Methods") are used to ensure the robustness of the result. In HIST, the ISM diabatic heating rate increases from lead of 1–2 days to lag of 0 days, thereby reaching a peak, and finally decays to retain weak positive heating from lag of 1–2 days to lag of 7–8 days corresponding to weakening of the Indian precipitation anomalies (Fig. 1). The life cycle of the abnormal ISM activities is approximately 3 pentads (Fig. 1 and Supplementary Figs. 1, 2). From lead of 1–2 days to lag of 1–2 days, the strong ISM diabatic heating triggers Rossby wave at 200-hPa that originate from the Indian subcontinent and terminate over East Asia, forming the CGT-like wavy pattern over the mid-latitudes of Eurasia (Fig. 1; see "Methods" for wave activity flux; hereafter, WAF). Coincident with the center of action of the CGT, East Asia is covered by an anticyclone at 200-hPa during these days, although the WAF later vanishes (Fig. 1; lag of 7–8 days). Meanwhile, a dipole-like precipitation pattern is found in East Asia, with a wet (dry) band in the north (south). The wet band over East Asia is intensified gradually after the lag0 day

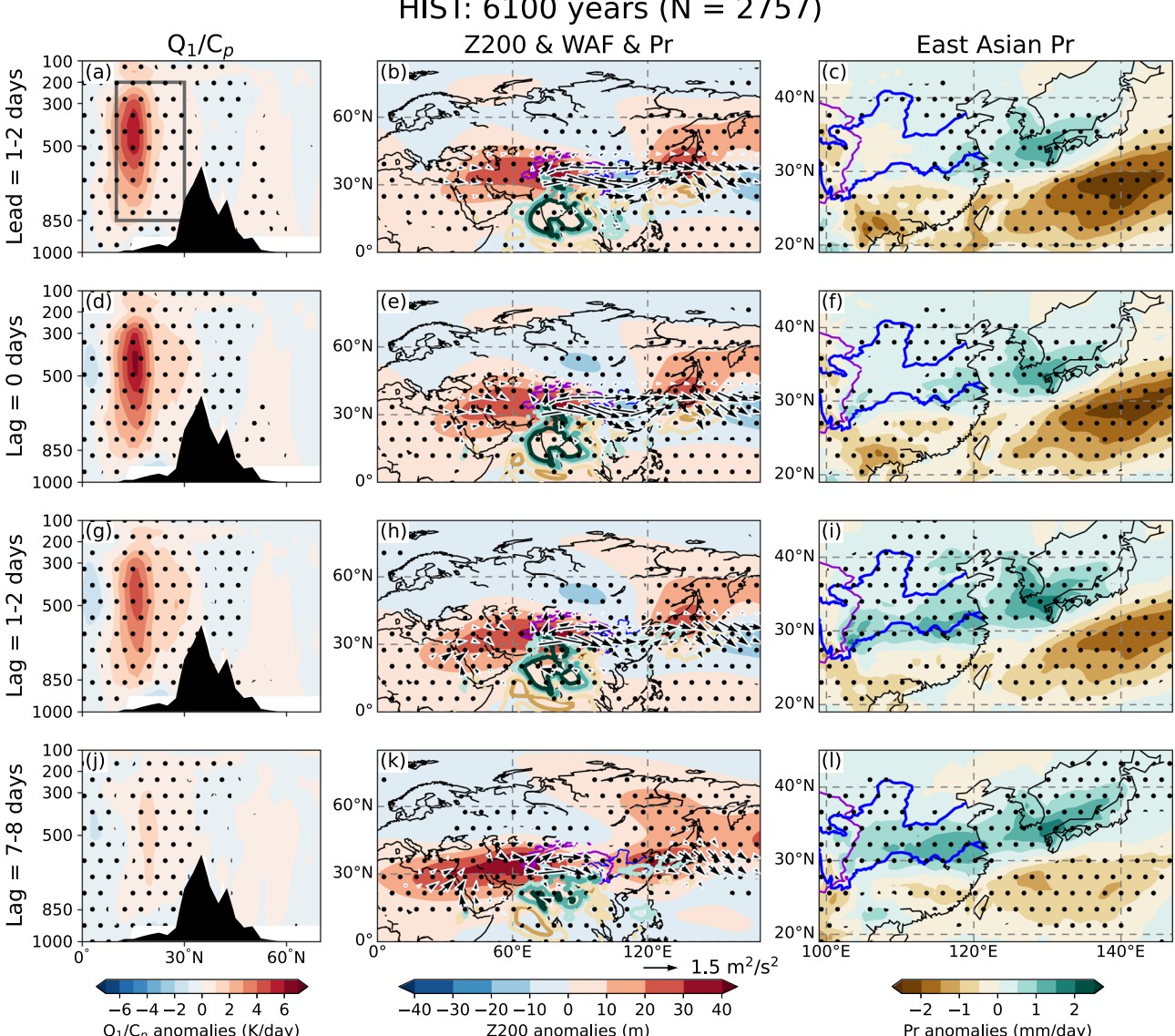

**Fig. 1 | Simulated lead–lag characteristics of circulation and precipitation relative to abnormal Indian summer monsoon heating.** Lead–lag composite at **a**–**c** lead of 1–2 days, **d**–**f** lag of 0 days, **g**–**i** lag of 1–2 days, and **j**–**l** lag of 7–8 days based on large ensemble historical climate experiment (HIST). **a**, **d**, **g**, **j** Cross-section of diabatic heating rate anomalies averaged over 70°–85°E, **b**, **e**, **h**, **k** 200-hPa geo-potential height anomalies (shading), wave activity flux (arrows), and precipitation anomalies (contours, unit: mm/day; with intervals of ±1, ±2, and ±4, positive and negative values are shown in green and brown contours, respectively), and **c**, **f**, **i**, **l** precipitation anomalies over East Asia. The number on the top (N = 2757) represents the number of samples used for the composite. Dotted areas denote anomalies significant at the 99% confidence level based on the two-sided Student's *t* test.

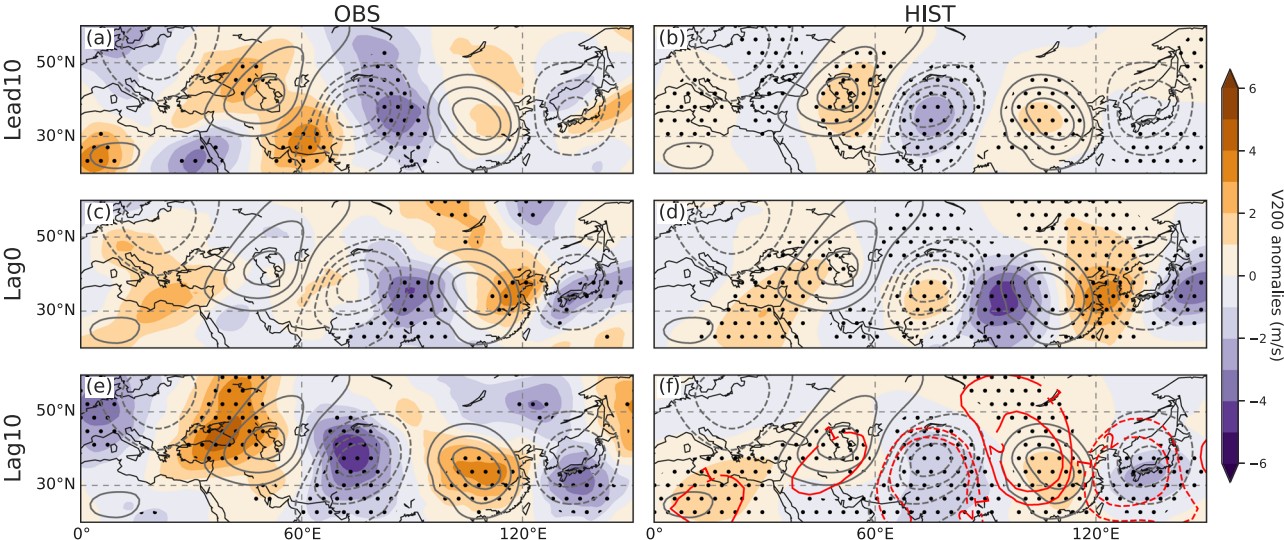

**Fig. 2 | Precedent and subsequent circumglobal teleconnection (CGT) wave train relative to abnormal Indian summer monsoon heating.** The shading indicates the composite 200-hPa meridional wind anomalies at **a** lead of 10 days, **c** lag of 0 days, and **e** lag of 10 days for observation and for large ensemble historical climate experiment (HIST; **b**, **d**, **f**). Red contours in (**f**) are from linear baroclinic model (LBM) experiment using June basic state, averaged from 30 to 50 days when LBM integration reached equilibrium. Gray contours show the eigenvectors of HIST first leading mode of interannual 200-hPa meridional wind during June to September (CGT pattern). Dotted regions denote anomalies significant at the 90% and 99% confidence level for the observation and HIST, respectively.

and it persists over nearly 2 pentads. The dry band, conversely, exists before the lag0 day and has small fluctuations over time. We therefore conjecture that the dry band may not be influenced by ISM heating (Fig. 1 and Supplementary Fig. 2). As for the observations, the result is almost identical to that of HIST, except that the wet band begins to form on the lag0 day and becomes increasingly evident over the following days (Supplementary Figs. 1, 2). Meanwhile, we anticipated that propagation of the summer intraseasonal oscillation might produce a similar result. However, we confirm that the dipole precipitation is not prominent when the 10–20-day or 30–60-day bandpass filter was applied.

The CGT is known as leading summer circulation mode over the mid-latitude Eurasia[25], but the relationship we obtained in June by the lead–lag composite is nonetheless absent in July to September. Sensitivity experiments using linear baroclinic model (LBM, see "Methods") also show that the ISM diabatic heating triggers a strong regional CGT wave train that propagates toward East Asia along the westerlies in June but not in July to September (Supplementary Fig. 3). The mechanism for the interaction among CGT, ISM, and EASM in June (i.e., Fig. 1 and Supplementary Fig. 1) is discussed in the next section.

## Role of CGT as a bridge between ISM and EASM

In this section we depict how CGT is able to bridge the ISM and the EASM in June. Two different types of CGT indices, CGTI–PC1 and CGTI–DW (see "Methods"), are calculated to track the subseasonal CGT variation, and their mean variations from lead of 20 days to lag of 20 days are shown in Supplementary Fig. 4a, b. They both show a bimodal M-shaped (i.e., peak-valley-peak) time variation, meaning CGT activity is strengthened in pre- and post-lag0 day. Power spectrum analysis for CGTI–PC1 shows a peak around 0.06–0.08 day$^{-1}$ which corresponds to 10–20-day, suggesting that CGT's main periodicity is close to the timescale of biweekly oscillation (Supplementary Fig. 4c, d). The significant power peak (i.e., exceeding the 95% red noise significance limit) for each selected event falls in the 5–25-day range (Supplementary Fig. 4e), with a mode at 0.07 day$^{-1}$ (-14-day, Supplementary Fig. 4f). These results suggest that the CGT behaves as a quasi-biweekly oscillation.

Figure 2 illustrates the time variations of the spatial pattern of CGT. During the first peak of the M-shaped CGT subseasonal variation, there is a wave train running from western Europe toward the ISM domain. At this stage, the intensity of the wave train is rather faint over East Asia (Fig. 2a, b). Near lag0 day, the vigorous ISM diabatic heating forces a local wave train to East Asia, but it is orthogonal to the CGT (Fig. 2c, d), as denoted by the valley of the M-shaped CGT subseasonal variation in Supplementary Fig. 4a, b. Few days later, the CGT emerges again but much more energetically in the East Asian region (Fig. 2e, f). According to the LBM experiment using June basic state, the ISM-induced wave train well overlaps the CGT wave train over East Asia on lag of 10 days, suggesting their local resonance along the westerlies (Fig. 2f). The transition of the wave pattern between lag of 0 days and lag of 10 days over East Asia is proposed to be linked to the forced response by ISM heating as a similar transition pattern was also seen in the LBM simulation.

Overall, the results suggest that CGT wave train interacting with ISM diabatic heating exhibits a character of quasi-biweekly oscillations. The first phase is the propagation of wave train from western Europe, far upstream of the ISM domain. The second phase is the onset of the ISM convection and diabatic heating which excites the local wave train and maintains propagation to East Asia. In this instance, the ISM heating activities might be related to European-CGT wave train propagation to ISM domain, as illustrated in the previous study[33,34]. Finally, the wave train takes on a CGT-like pattern as it extends horizontally along the westerly jet. One may notice this condition leads to the intensification of southerly winds over East Asia. The role of ISM-induced wave train and the southerly winds on East Asian precipitation will be elaborated in the next section.

## Clustering for jet stream pattern

In this section, the interaction of CGT and wet band is analyzed based on their temporal evolution following the emergence of the Indian heating (i.e., event0 day; see "Methods"). In the previous section, we considered the ISM diabatic heating as essential forcing to excite the Rossby wave that might interact with the CGT. Earlier studies suggested that the CGT can be triggered by forcings other than ISM heating, such as diabatic heating near the Mediterranean[25], land

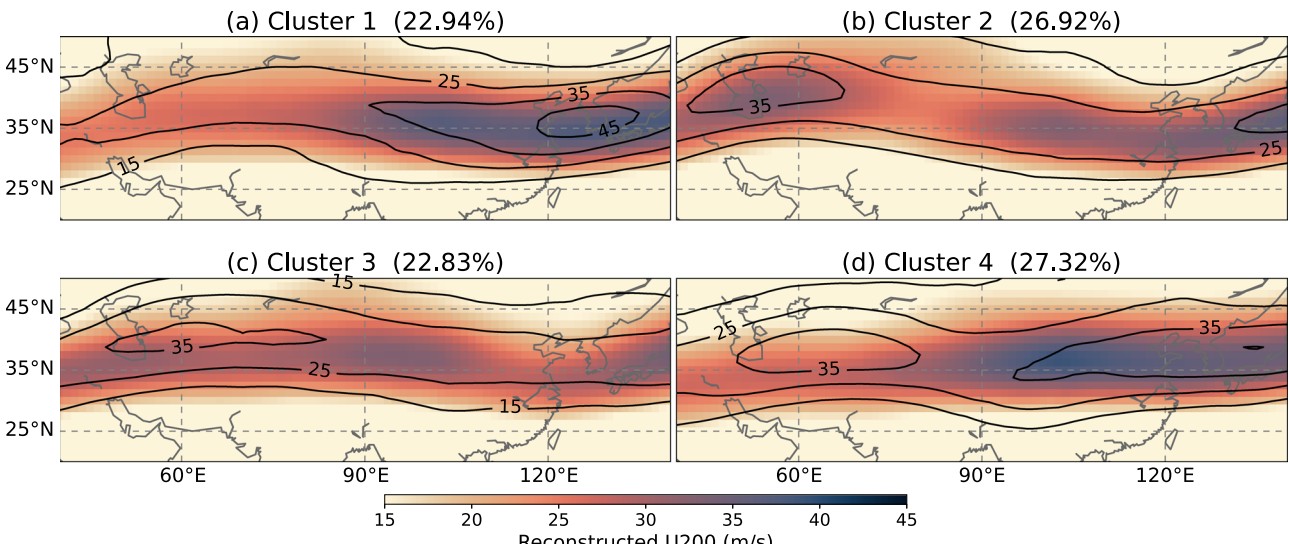

**Fig. 3 | Clustered jet stream patterns.** The reconstructed 200-hPa zonal wind (jet stream) patterns for **a** cluster 1, **b** cluster 2, **c** cluster 3, and **d** cluster 4 created by K-means clustering. Color shading represents values from large ensemble historical climate experiment (HIST), and black contours represent values from JRA-55 reanalysis. The value in the parenthesis denotes the occurrence frequency obtained from all 2786 samples (i.e., 2757 samples from HIST and 29 samples from JRA-55 reanalysis).

surface processes over the Middle East[35], jet stream instability, and heating related to tropical convection[36]. It is widely accepted that the subtropical jet could determine the propagation of the CGT as a waveguide. Here, K-means clustering is applied to the reconstructed jet stream fields at event0 day to elucidate how ISM forcing might propagate along the waveguide and stimulate the East Asian rainband (see "Methods"). Clusters 1 and 4 exhibit a strong jet core over East Asia, while clusters 2 and 3 show a wavy jet stream with deacceleration near East Asia in comparison with clusters 1 and 4 (Fig. 3). The clustered East Asian jet patterns are also found related to the jet meridional displacement, i.e., in clusters 1 and 4 the East Asian jets are displaced to north, with corresponding north-positive and south-negative zonal wind anomalies extending from the upper troposphere to near surface (Supplementary Fig. 5). In contrast, in clusters 2 and 3, the jets are displaced to south with associated reversed zonal wind anomalies in the troposphere. The following discussion is mainly based on the HIST owing to sufficient samples.

Similar to Fig. 1, lead–lag composite analyses are conducted here but for each of the cluster. At lag of 3–4 days, all clusters exhibit a dipole precipitation pattern over East Asia (Fig. 4e–h). The wet band is shown to pre-exist in clusters 1 and 4 and is intensified after the lag0 day (Fig. 4i–l). Conversely, for clusters 2 and 3, the wet band could be activated by ISM heating. The wet band might be related to ISM heating activities, as illustrated in Fig. 1 and Supplementary Figs. 1, 2. The strength of ISMH in each cluster is nearly equivalent at lag0 day (-2.20 K/day; Fig. 4a–d). Therefore, the difference in the atmospheric fields among the clusters is likely attributable to the temporal evolution of the dynamical processes that include interaction between the subtropical jet and ISM heating, as well as the effects from other remote forcings that could be captured by the jet. Strong ISMH activity is connected to anomalous 200-hPa CGT-like circulation patterns in all clusters at lag0 day but with slight differences (Fig. 4e–h). The CGT in clusters 1 and 4, shows a strong cold trough located near Mongolia and extending to the north of the Yangtze River Basin, whereas the other clusters show troughs located near the Yangtze River Basin. These differences can be attributed to differential westerly jet stream pattern among clusters. The vigorous ISM diabatic heating near lag0 day perturbs the air in proximity to the westerly jet, serving as a Rossby wave source. The westerly jet stream acts as a waveguide[37], which could confine the wave and direct it toward East Asia. Our findings

indicate that clusters 2 and 3 exhibit a southward bending of the jet stream over East Asia (i.e., East Asian jet displaced to south), resulting in a southward motion of the associated wave train and the formation of an anomalous cold center (negative geopotential height anomaly) over East Asian regions on lag0 day (Fig. 4q–t and Supplementary Fig. 6). The cold center is consequently guided southward by the jet (south of 35°N). Conversely, clusters 1 and 4 demonstrate a relatively straight jet stream (i.e., East Asian jet displaced to north), resulting in a predominantly zonal propagation of the wave toward East Asia. This dynamical constraint leads to the cold anomaly center being positioned further north in the East Asian sector (Supplementary Fig. 6a, d). The waveguide effect is also confirmed by stationary Rossby wavenumber ($K_s$)[37], which is calculated based on the jet pattern of Fig. 3. The dominant wavenumbers are longer in clusters 2 and 3 while shorter in clusters 1 and 4 (Supplementary Fig. 7). Hence, the CGT pattern and the location of the downstream trough differ between clusters (Fig. 4e–h).

East Asia is affected by anomalous upper-level southerly winds and associated 500-hPa anomalous warm advection during lag days, regardless of the differences in the location and strength of the cold trough between clusters at lag0 day (Fig. 4i–l, q–t). These upper-level southerly winds cause warm advection over East Asia (Fig. 4m–p). Namely, for all clusters, strong anomalous 500-hPa warm advection located near the wet band area, the main body of which is overlapped by the anomalous southerly wind-driven warm advection during lag days (Supplementary Fig. 8), are possibly related to ISM diabatic heating. Specifically, for clusters 2 and 3, anomalous warm advection at 500-hPa appear over East Asia a few days earlier than the occurrence of the wet band. Meanwhile, for clusters 1 and 4, intensification of the wet band is confirmed after the excitation of ISMH peak, accompanied by a strengthened anomalous 500-hPa warm advection, although the anomalies were already present at lead of 10 days. The pre-existing warm advection in clusters 1 and 4 is presumably due to the prevailing southerly winds. We hypothesize that the jet meridional displacement in East Asia sector is a sign of the pre-existing southerly winds and wet band. This speculation is supported by the subseasonal lead–lag correlations, which suggest the occurrence of southerly winds prior than wet band precipitation, while jet meridional displacement tends to occur after the appearance of both southerly winds and wet band precipitation (Supplementary Fig. 9).

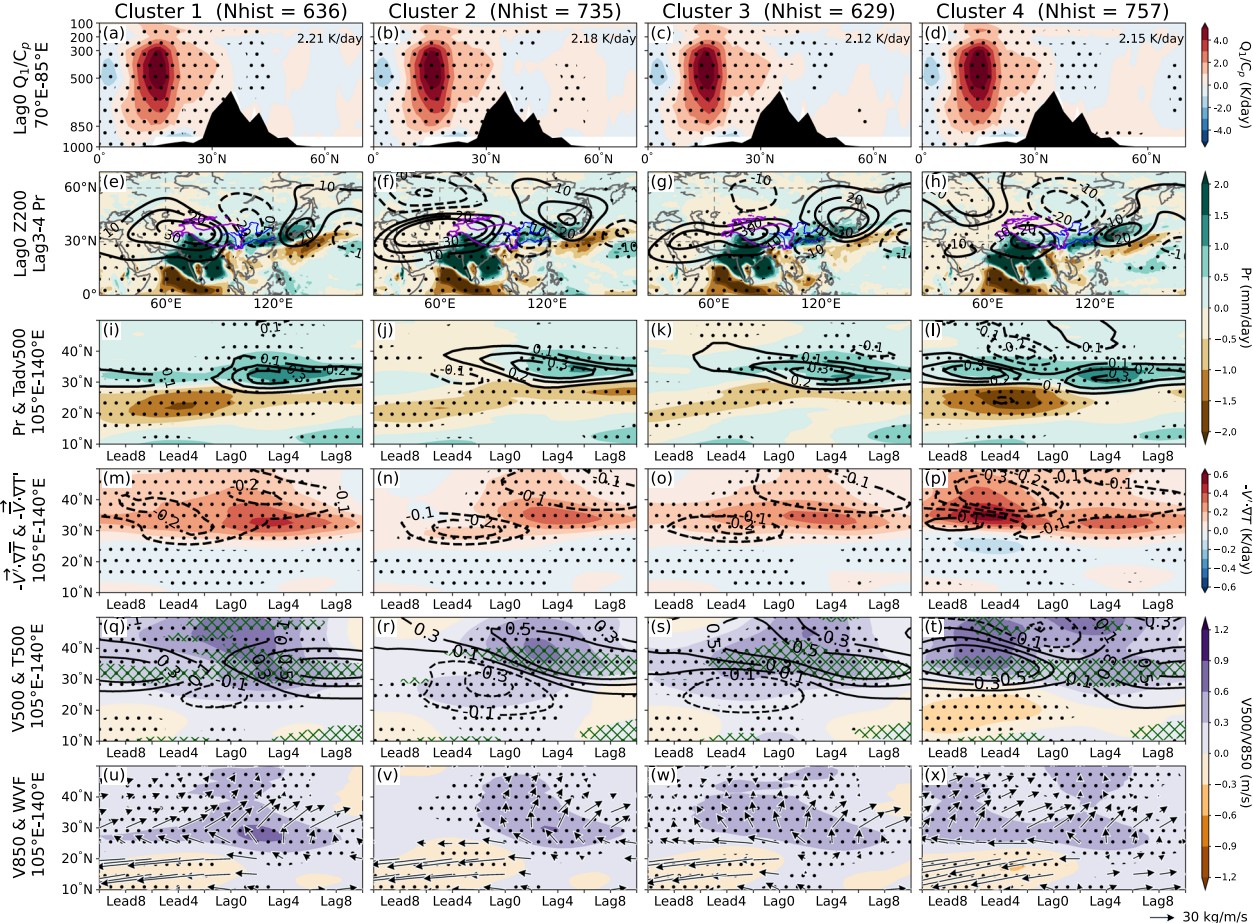

**Fig. 4 | Simulated lead–lag dynamical characteristics relative to abnormal Indian summer monsoon (ISM) heating in each cluster. a–d** Composite of diabatic heating rate anomalies averaged over 70°–85°E at lag of 0 days, where 850–200 hPa mass-weighted and 10°–30°N averaged values (i.e., ISMH) are displayed on upper right, and **e–h** composite of precipitation anomalies (shading) and 200-hPa geopotential height anomalies (contours unit: m) at lag of 3–4 days. Temporal evolutions of the zonal mean fields averaged over 105°–140°E computed from lead of 10 days to lag of 10 days of **i–l** zonal mean precipitation (shading) and 500-hPa temperature advection (contours, unit: K/day) anomalies, **m–p** 500-hPa temperature advection anomalies computed by anomalous wind field with climatological temperature gradient (shading) and climatological wind field with anomalous temperature gradient (contours, unit: K/day), **q–t** 500-hPa meridional wind (shading) and temperature (contours, unit: K) anomalies, and **u–x** 850-hPa meridional wind (shading) and vertical integrated water vapor flux (arrows) anomalies. The green hatching in (**q–t**) indicates vertical velocity (unit: Pa/s) lower than 0, significant at the 99% confidence level based on the two-sided Student's *t* test. **a, e, i, m, q, u** cluster 1, **b, f, j, n, r, v** cluster 2, **c, g, k, o, s, w** cluster 3, and **d, h, l, p, t, x** cluster 4. The stippled regions denote anomalies significant at the 99% confidence level based on the two-sided Student's *t* test. The number of samples used for the composite for each cluster is shown on the top (Nhist).

These results reveal that CGT wave train steers anomalous warm advection and ascending air over East Asia in lag days (Fig. 4i–l contours, and Fig. 4q–t green hatching). The warm advection and associated upward motion lead to greater precipitation that can anchor the EASM rainband, consistent with the findings of previous study[38]. Meanwhile, the location of the cold center guided by the westerlies jet is also important since it has an effect to strengthen anomalous cold advection to East Asia and determines the sign of the temperature advection anomalies (Fig. 4m–p). The low-level water vapor, in addition, fuels the wet band in response to the upper-level anomalous southerly winds. The maximum 500-hPa meridional wind anomalies are located to the north of 30°N, while those at 850 hPa are generally to the south of 30°N (Fig. 4q–t, u–x). This indicates that the southerly winds over East Asia tilts northward with height in the troposphere. Thus, interaction between 500-hPa warm advection and 850-hPa water vapor flux driven by the southerly winds is essential for the growth of the wet band. In summary, our results suggest that the CGT wave train tends to steer mid-tropospheric warm advection and lower-tropospheric water vapor toward East Asia that contribute to the formation of the wet band. Moreover, we infer that ISM heating acts as a trigger (for clusters 2 and 3) of the wet band, but also as a reinforcer (for clusters 1 and 4) if abnormal southerly winds and associated warm advection are pre-existing. Meanwhile, the location and intensity of the cold center in part of the CGT guided by the jet is also shown to be important for the wet band prediction.

The WAF for each cluster is presented in Fig. 5 to explore the pathway of wave propagation. In cluster 1, the CGT is pre-existing before ISM onset, which is consistent with the pre-existent wet band. Centers with positive stream function anomalies can be identified over the North Atlantic Ocean, western Mediterranean, northern Indian subcontinent, and East Asia in lead days. The role of the ISM in this case is to maintain the CGT and associated wet band over East Asia. In clusters 2 and 3, the WAF originates from the Ural Mountains and moves toward the Indian subcontinent in lead days. A strong meridional Rossby wave is a possible cause of the meandering of the jet stream (Fig. 3) and the occurrence of ISM heating activities[33,34], indicating potential interaction between the polar jet and the subtropical jet. In this case, ISM heating is the major trigger of the abnormal warm advection over East Asia and the associated wet band. In cluster 4, wave activity is seen at high latitudes prior to ISM diabatic heating (i.e., in lead days), whose pathway originates from western Europe and runs across the Ural Mountains and East Asia, causing an anomalous

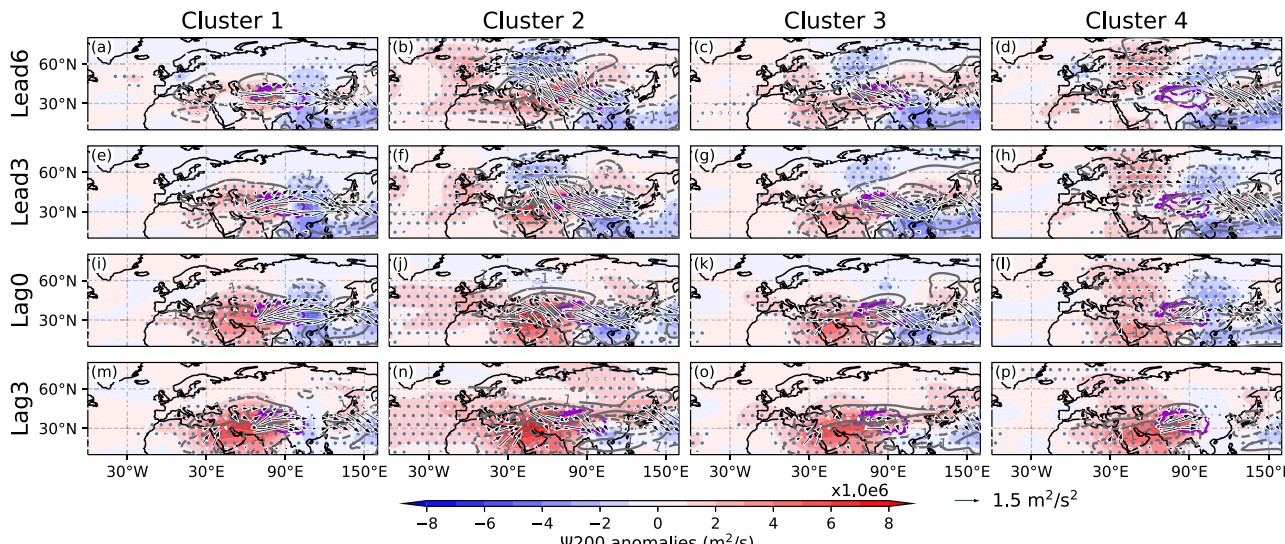

**Fig. 5 | Lead–lag composite of wave activities relative to abnormal Indian summer monsoon heating in each cluster.** Stream function (color shading), 200-hPa zonal wind anomalies (contours, unit: m/s), and wave activity flux (arrows) for **a**–**d** lead of 6 days, **e**–**h** lead of 3 days, **i**–**l** lag of 0 days, and **m**–**p** lag of 3 days. **a**, **e**, **i**, **m** cluster 1, **b**, **f**, **j**, **n** cluster 2, **c**, **g**, **k**, **o** cluster 3, and **d**, **h**, **l**, **p** cluster 4. The stippled regions denote anomalies significant at the 99% confidence level based on the two-sided Student's $t$ test. Wave activity flux with values lower than 0.01 are omitted.

anticyclone over East Asia. On this occasion, the role of ISM heating is to reinforce warm advection over East Asia, similar to cluster 1, and to help maintain the wet band over East Asia. Our results demonstrate that the dipole precipitation pattern over East Asia is influenced by various remote factors with different pathways. ISM heating encourages development of the wet band over East Asia in collaboration with these remote factors. In subseasonal lead–lag composite, we notice discrepancy in the precipitation response to ISM heating between the observations and HIST (Supplementary Fig. 1 and Fig. 1). The pre-existing wet band precipitation anomalies in HIST is dissimilar to the result of the observations (Supplementary Fig. 2). It is interpreted that the greater number of samples in HIST could detect cases in which ISM heating reinforces the pre-existing rainband.

## Discussion

In this study, we focus on the ISM–EASM connection on the subseasonal timescale, building upon the observations and large-ensemble experiment. Here we provide the quantitative insights regarding the contribution of ISM to the EASM wet band on the subseasonal timescale. Figure 6 compares probability distribution of daily wet band precipitation anomalies created using the entire study period (i.e., all days from June and first 2 pentads of July) and using only lag of 0–10 days as the period that potentially contains the ISM influence. In this figure, positive precipitation anomalies greater than 0.5 mm/day is regarded as wet band events. It is clear that the wet band precipitation anomalies during lag of 0–10 days tend to shift to positive values, with mean wet band precipitation anomalies of 1.04 mm/day for observations and 0.66 mm/day for HIST. Among all wet band events, the probability of wet band events from June to early July with presence of lag of 0–10 days is 21.02% for observations and 16.72% for HIST. Noteworthy, these probabilities rise to 23.26% and 19.77%, and further to 31.58% and 24.67% for the events exceeding +1 and +2 standard deviations of wet band precipitation anomalies in the observation and HIST, respectively. Therefore, the probability of wet band events with presence of lag days (i.e., influenced by the abnormal ISM heating) increases with increasing precipitation anomalies (Fig. 6). These findings suggest that the ISM plays a more prominent role in stronger wet band precipitation anomalies. In particular, the exceedance probabilities of +1 standard deviation of wet band precipitation anomalies

during lag of 0–10 days are 1.83 times higher in observations and 1.59 times higher in HIST, respectively, in comparison to the probability estimated for daily wet band precipitation anomalies in June and early July. Therefore, this indicates the likelihood of wet band event is higher when abnormal ISM heating exists.

We are hereby also interested in if the subseasonal processes could be beneficial for the prediction of June mean precipitation, because the leading mode of June monthly mean precipitation in East Asia also exhibits a similar dipole pattern in both GPCP[39] and HIST (Supplementary Fig. 10a, e). We select cases when these criteria on different time scales are either partially satisfied or both satisfied based on the detection of lag0 day, meaning the active ISM heating, and the principal component for interannual June precipitation (hereafter, EAPR-PC1) which shows dipole precipitation pattern. The lag0 day was targeted for the case that occurred only within the first 5 pentads in June, considering the lead–lag relationship between the ISM and the EASM (Fig. 1). Comparison of monthly precipitation in June with and without detection of lag0 day (i.e., strong and persistent ISMH) allows us to establish the impact of subseasonal ISM heating on monthly East Asian precipitation.

The EAPR-PC1 high years (defined as EAPR-PC1 > 0.5) with detection of lag0 day show strong connection between Indian precipitation and the East Asian dipole precipitation pattern (Supplementary Fig. 10b, f) in June. Meanwhile, the EAPR-PC1 high years without strong ISMH show the East Asian dipole pattern of precipitation, but Indian precipitation is close to neutral (Supplementary Fig. 10c, g). Given the complex influencing factors of the EASM, the dipole precipitation pattern could also be induced by factors other than the ISM, including the Pacific–Japan pattern-like feature evident in the observational data shown in Supplementary Fig. 10c. Mean precipitation in June for the year with lag0 day shows a relatively weak dipole pattern in the result of the observations and HIST (Supplementary Fig. 10d, h), whose projected EAPR-PC1 scores are 0.65 and 0.37, respectively. These scores represent the extent to which subseasonal ISM affects interannual EASM. Meanwhile, for years where lag0 day was successfully detected, the EAPR-PC1 score shifts accordingly to the positive values. We also confirm that higher ISMH values (indicating stronger ISM forcing) correspond to higher interannual EAPR-PC1 scores, as expected (Supplementary Fig. 10i, j).

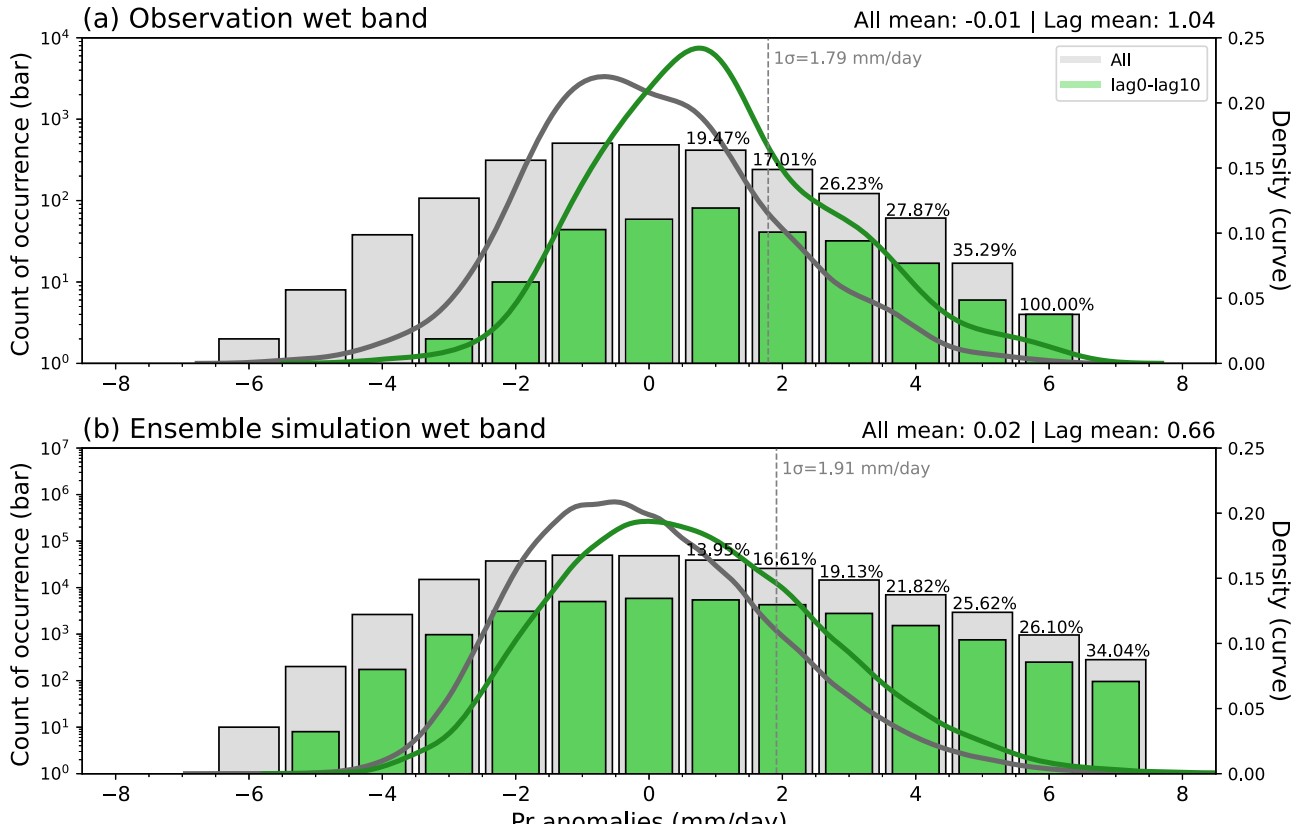

**Fig. 6 | Potential contribution of Indian summer monsoon heating to East Asian precipitation at subseasonal timescale.** The probability distribution of wet band precipitation anomalies based on **a** observation and **b** large ensemble historical climate experiment (HIST). Gray and green colors are for all days of the entire study period (i.e., days from June and first 2 pentads in July) and for days during lag of 0–10 days, respectively. The bars denote the count of occurrences (left Y-axis) for every 1 mm/day bin, and percentage value at the top of each bar refers to wet band occurrences coinciding with lag of 0–10 days. The occurrence of the wet band is defined as positive precipitation anomaly greater than 0.5 mm/day. The domain for the wet band is depicted as the green rectangle in Supplementary Fig. 1. The value of +1 standard deviation of June to early July wet band precipitation anomalies is illustrated by dashed vertical line. The curves represent the probability density (right Y-axis), with the mean values at the top right of each panel.

These findings underscore the role of subseasonal ISM heating on the formation of the East Asian rainband. The subseasonal ISM heating signal in the interannual EASM is magnified or obscured if multiple processes coexist. The unstable relationship of monthly or seasonal precipitation between the ISM and EASM, as mentioned in introduction, is therefore speculated to indicate that the monthly average precipitation does not sufficiently represent the physical ISM–EASM connection signal.

We found a robust connection between the EASM and ISM in early summer. Our subseasonal analysis targeting this relationship reveals that ISM diabatic heating can trigger a CGT wave train that steers warm advection toward East Asia. The anomalous warm temperature advection, in conjunction with low-level moisture transport driven by the southerly winds in East Asia as a part of the CGT, induces greater precipitation anomalies over central East China, the Korean Peninsula, and southern Japan. The findings of this work provide a basis for ISM–EASM connection on the subseasonal timescale. More importantly, the large-ensemble simulation enables comparison of the temporal evolution of the atmospheric fields among various jet stream features, thereby enhancing understanding of the hydrological cycle in East Asia and benefitting subseasonal forecasting of heavy rainfall of the EASM. The present study leaves certain issues that must be understood in the future; for instance, further studies will be needed on the factors that control the dry band over East Asia (some clues are shown that it is related to low-level easterly winds before lag0 day, Fig. 4u–x), and the processes behind ISM heating including how its timespan from initiation to peak is determined.

## Methods

### Observational and reanalysis data

The observed leading precipitation variability mode was investigated using monthly data from 1979 to 2020. The monthly precipitation data (2.5° × 2.5°) from the Global Precipitation Climatology Project (GPCP)[39] version 2.3 was used (Supplementary Fig. 10; EAPR–PC1).

The subseasonal analysis was conducted over an extended period (1958–2020) to enlarge the sample size of episodes. The daily mean atmospheric circulation and hydrological parameters were derived from the Japanese 55-year Reanalysis datasets[40] (JRA-55) with spatial resolution of 1.25° × 1.25°. The analysis also used Asian Precipitation–Highly Resolved Observational Data Integration Towards Evaluation of Water Resources (APHRODITE) version 2[41] daily data for the period 1958–2015.

### Simulation data

A large-ensemble atmospheric global climate model experiment was used to obtain a suitable number of cases to satisfy the requirements for analysis on subseasonal atmospheric variability. The output from the 100-member ensemble simulations was obtained from the "Database for Policy Decision-Making for Future Climate Change" (d4PDF)[42]. We used the historical climate experiment (HIST) between 1951 and 2011, with horizontal mesh size of 60 km, and the circulation-related output, with 1.25° × 1.25° resolution. This experiment was forced using observed monthly mean SST and sea ice from the COBE-SST2[43] dataset and other observed external forcing agencies such as greenhouse gases, aerosols, and ozone. The ensemble members were created using

perturbed lower boundary conditions and initial conditions. The monthly mean and daily mean output from HIST were analyzed in this study.

## Diagnostic approaches

The apparent heat source[44] ($Q_1$) defined below was computed based on daily fields to diagnose the thermodynamic processes:

$$Q_1 = C_p \left(\frac{p}{p_0}\right)^{\frac{R}{C_p}} \left(\frac{\partial \theta}{\partial t} + \vec{V} \cdot \nabla \theta + \omega \frac{\partial \theta}{\partial p}\right) \tag{1}$$

where $\theta$ is potential temperature, $\vec{V}$ is the velocity of the horizontal wind, $\omega$ is the vertical pressure velocity, $R$ stands for the gas constant, $C_p$ is the specific heat at constant pressure for dry air, and $p$ is pressure ($p_0 = 1000$ hPa).

The wave activity flux[45] (WAF) defined as below is calculated to diagnose the stationary Rossby wave propagation:

$$\vec{W} = \frac{p \cos \varphi}{2|\vec{V}|} \begin{pmatrix} \frac{\bar{u}}{a^2 \cos^2 \Phi}\left[\left(\frac{\partial \psi'}{\partial \lambda}\right)^2 - \psi'\frac{\partial^2 \psi'}{\partial \lambda^2}\right] + \frac{\bar{v}}{a^2 \cos \Phi}\left[\frac{\partial \psi'}{\partial \lambda}\frac{\partial \psi'}{\partial \Phi} - \psi'\frac{\partial^2 \psi'}{\partial \lambda \partial \Phi}\right] \\ \frac{\bar{u}}{a^2 \cos \Phi}\left[\frac{\partial \psi'}{\partial \lambda}\frac{\partial \psi'}{\partial \Phi} - \psi'\frac{\partial^2 \psi'}{\partial \lambda \partial \Phi}\right] + \frac{\bar{v}}{a^2}\left[\left(\frac{\partial \psi'}{\partial \Phi}\right)^2 - \psi'\frac{\partial^2 \psi'}{\partial \Phi^2}\right] \end{pmatrix} \tag{2}$$

where $p$ is pressure, $\bar{u}$ and $\bar{v}$ are the basic state of the horizontal winds, set as climatological monthly mean values in this study, $|\vec{V}|$ denotes the wind velocity computed from $\bar{u}$ and $\bar{v}$, $\Phi$ and $\lambda$ are latitude and longitude, respectively, and $\Psi'$ represents anomalies of the stream function.

The CGT indices are calculated to measure the subseasonal variations of the CGT. The CGT reference pattern is created by applying EOF analysis to seasonal mean (June to September) meridional wind at 200-hPa over the domain of 20°–60°N, 0°–150°E, following previous study[25]. The leading patterns from JRA-55 and HIST resemble each other (Supplementary Fig. 11). Hence, we used HIST leading pattern as reference for convenience. Then, daily June 200-hPa meridional wind anomalies are projected onto this reference pattern to obtain the daily CGT index (CGTI–PC1) for both JRA-55 and HIST, which represents the subseasonal variation of CGT wave pattern. Another CGT index[22] was also computed for comparison, by area-averaging the 200-hPa geopotential height anomalies over 35°–40°N, 60°–70°E (CGTI–DW).

The anomalies in this study represent the deviation of the variables from their 11-year moving average of the same calendar month or Julian day according to the time scales of the data to remove the decadal variability. Thereafter, an additional 5-day moving average was applied to the daily anomaly fields.

## Selection of strong Indian monsoonal diabatic heating events

The abnormal Indian monsoonal diabatic heating events of June were collated based on the following criteria for both JRA-55 and HIST. The daily anomalies of the diabatic heating rate ($Q_1/C_p$; unit: K/day) averaged over 10°–30°N, 70°–85°E were computed for each year from May 1 to July 31. The definition of the anomaly follows that in previous section. Subsequently, the vertical mass-weighted mean from 850 to 200 hPa gives the averaged ISM heating released to the troposphere. Because 1 standard deviation of daily ISMH in June for JRA-55 and HIST is 1.06 and 1.10 K/day, respectively, an abnormal ISM heating event was defined as the case when ISMH is greater than 1.00 K/day with duration of 5 or more consecutive days. The first day of each event is referred to as the "event0" day, and the day observing the highest ISMH value during each event is designated as the "lag0" day (Supplementary Fig. 12). To highlight the event in June, the lag0 day must occur in June; otherwise, it is rejected. If multiple events are detected in a year, the event with the highest ISMH (i.e., the strongest heating anomaly) is adopted. Overall, 29 lag0 day and 2757 lag0 day were selected based

on JRA-55 and HIST, respectively. Finally, composite analysis was performed to investigate the lead–lag relationship relative to the lag0 day.

The ISMH index we defined is a diagnosis capable to represent the gradual evolution of the ISM heating corresponding to large-scale circulation. To illustrate the relationship between ISMH and ISM precipitation, the June ISM precipitation index was calculated from the same domain of ISMH (10°–30°N, 70°–85°E). Their correlation coefficients are 0.80 and 0.99 (both with $p < 0.01$), respectively, for APHRODITE precipitation versus JRA-55 ISMH and HIST precipitation versus HIST ISMH. This suggests that ISMH can serve as a proxy of ISM precipitation, especially considering the limited availability of daily in situ precipitation records for the observational period of this study, which spanned from 1958 to 2020.

## Linear baroclinic model experiments

To investigate whether the ISM could trigger East Asian CGT, we conducted experiments using the linear baroclinic model (LBM)[46,47], which is feasible to simulate a linear response to a prescribed forcing. The LBM was set to the resolution of T42L20 (128 × 64 grids horizontally and 20 sigma levels vertically). Horizontal diffusion with an e-folding timescale of 2 h was applied. We considered five scenarios, representing each month from May to September as basic states, using JRA-55 monthly mean climatological values from 1979 to 2020. For each scenario, we imposed the same composite June lag0 day ISM diabatic heating from observation (Supplementary Fig. 1) over the 10°–30°N, 70°–85°E region, which was regridded from JRA-55 to T42L20 resolution (Supplementary Fig. 13).

## K-means clustering

To examine how ISM diabatic heating interacts with the preconditioning waveguide and its potential influence on the ISM–EASM connection, i.e., the jet patterns were classified. The jet axes of event0 day obtained from both JRA-55 and HIST were input together in the K-means clustering[48]. The methodology to obtain the jet axis was as follows. First, zonal wind at 200-hPa (U200) was reconstructed by adding U200 anomalies at the event0 day and their corresponding pentad (±2-day) climatological U200 (Supplementary Fig. 14a, b). Second, the jet axis was identified as the latitudinal location of the maximum reconstructed U200 at each longitude within the range of 20°–50°N. This domain setting was intended to avoid potential influence by the polar front jet that appears in higher latitudes[49]. The input vectors to the K-means clustering included the values of latitude and longitude of the jet and the reconstructed U200 along the jet axis over the sector 40°–140°E (Supplementary Fig. 14c; black box). Finally, we created four clusters of the jet pattern at the event0 day as the optimal configuration based on the "elbow method." We also input the same data into principal component analysis, and result indicates the population mean characteristics of four clusters are likely to be orthogonal to each other (Supplementary Fig. 15).

The mean duration between event0 day and lag0 day is about 4 days for each cluster (Supplementary Fig. 16). Therefore, event0 day is roughly corresponds to 4 days before lag0 day (i.e., lead of 4 days). We have confirmed that the cluster composite features derived from lead of 4 days are similar to those in event0 day.

## Data availability

The GPCP V2.3 data were obtained from http://gpcp.umd.edu. The APHRODITE data were downloaded from http://aphrodite.st.hirosaki-u.ac.jp/download/. The JRA-55 reanalysis data and d4PDF data used in this study are available in Data Integration and Analysis System (https://diasjp.net/en/dias-datasetlist/). The LBM source code can be requested via https://ccsr.aori.u-tokyo.ac.jp/~lbm/sub/lbm.html. The LBM related output in this study is available at: https://zenodo.org/record/7919668#.ZF17ZerP1D8.

## Code availability

The data in this study were analyzed with Python. Contact S.L. for specific code requests.

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

## Acknowledgements

S.L. was supported by JST SPRING, Grant Number JPMJSP2119. T.S. was supported by the Arctic Challenge for Sustainability II (ArCS II) program, Grant Number JPMXD1420318865, and SENTAN program, Grant Number JPMXD0722680734, funded by MEXT (Japan); and KAKENHI, Grant Nos. JP18KK0098 and JP19H05666, funded by the Japan Society for the Promotion of Science (JSPS). The authors thank Prof. Takeshi Horinouchi of Hokkaido University and Dr. Rakesh Teja Konduru of RIKEN Center for Computational Science for their comments. The authors also thank James Buxton MSc, from Edanz (https://jp.edanz.com/ac), for editing a draft of this manuscript.

## Author contributions

S.L. and T.S. designed the research project and methods. S.L. conducted the analyses with the help of T.N. and W.G. All authors (S.L., T.S., T.N., and W.G.) contributed to the preparation of the manuscript.

## Competing interests

The authors declare no competing interests.
