## [Peer Review File · Nature Communications]

East Asian summer rainfall stimulated by subseasonal Indian monsoonal heatingREVIEWER COMMENTS

Reviewer #1 (Remarks to the Author):

The authors attempt to find out whether the Indian summer monsoon (ISM) can serve as a predictor of the East Asian summer monsoon (EASM) on the subseasonal time scale in June. They argue that Indian subseasonal heating in June could enhance the East Asian rain band by triggering Rossby wave propagation along the subtropical jet and generating an anticyclone over the EA, which strengthens mid-tropospheric warm advection and lower-tropospheric moisture advection conducive to the EA rainband formation.

Overall assessment

The main conclusion of this work is flawed. The authors claimed that the Indian subseasonal heating in June is the critical factor controlling the dipole precipitation over EA. The authors presented Fig. 4 to support their claim. However, Figure 4 indicates the opposite: EA rainfall occurs before the ISM heating peak, and when ISM heating decays, the EASM rainfall intensifies, meaning that ISM heating is not forcing EA rainfall.

The presentation is illogical. Section 2.1 (dominant mode of June East Asian precipitation and its controlling factor) presented three figures (Figs. 1, 2, and 3). They all show interannual variations of the June precipitation anomaly, associated water vapor fluxes, and diabatic heat sources. These results are well-known in many previous works and are nothing new. Besides, the three figures' results have nothing to do with the central theme (subseasonal variation). What caused the inter-related variations was never discussed. Yet, the authors jumped to conclude, "The potential role of the ISM on the dipole precipitation pattern over East Asia is confirmed in the monthly mean field". (Line 129-130).

The citation of the literature in the introduction is ad hoc and does not acknowledge original contributions, reflecting an insufficient depth of the Authors' understanding and knowledge relevant to their study.

There are many other issues. However, the above concern makes me feel negative about the work. I recommend the rejection of the manuscript for publication.

Reviewer #2 (Remarks to the Author):

This is an interesting analysis with a primary emphasis on the role of the CGT in linking the ISM and ESAM on subseasonal time scales. According to their analysis and results, the authors suggested that diabatic heating anomalies associated with the ISM is critical to drive a regional component of the CGT over East Asia in June and synchronize rainfall variability along the path of the CGT from central India to East Asia. Actually, this physical connection and the role of the ISM in the formation of the CGT have been known for decades. The novelty of this new study primarily lies in its use of the K-mean clustering analysis and a large-ensemble simulation to reveal the connection of the two submonsoon systems over shorter time scales. However, a number of minor and major issues that are related to technical and scientific components of the study remain unclear and thus need to be addressed or clarified first before the manuscript can be considered for publication in the journal. Although the points I raised below regard many fundamental aspects of the formation of the CGT, I still feel that the study has strong potential to become an important piece of work that will greatly improve our physical understanding of the CGT.

1. The CGT is the leading NH circulation mode prevailing from June to September. The current study only focused on its variability in June. It is not clear to me how to apply what we learn from this month's situation to variations over other months.

2. In Fig1, the authors calculated the correlation of the blue and black curves and added some relevant discussion on this poor relationship. I don't fully understand why the authors took some

effort to do this calculation since it is not expected to see a high correlation of these two time series given that these two curves are generated in very different climate scenarios.

3. It is still not clear to me what we can learn from the clustering analysis. If I understand the method right, these four modes are not orthogonal to each other and there are some common features shared by these four modes. If so (they look very similar in Fig. 6 & 7), I feel the author should put more attention on these common features and also distinctions of each mode from the common features.

4. The study emphasized that the CGT mode presented here varies energetically on subseasonal time scales, using a lead-lag composite method covering data from day-10 to day+10. In the paper, it is better to further illustrate what the main periodicity of this feature is. A power spectrum analysis of the time series related to this subseasonal variability may help the authors answer this question.

5. Some minor wording issues pop up here or there in the current form. I prefer to place my main focus on these wording problems next round after these scientific issues I just raised could be ironed out.

Reviewer #3 (Remarks to the Author):

Comment on “East Asian summer rainfall stimulated by subseasonal Indian monsoonal heating” by Li et al.

The authors found that on the sub-seasonal timescales, an Indian anomalous diabatic heating may serve as a predictor for East Asian rainfall in early summer. The authors first identified the leading mode of June rainfall in East Asia on the interannual timescales and revealed the relation to Indian rainfall. Then, the authors proposed that the Indian diabatic heating anomaly can lead to East Asian rainfall variability through a wave train along the Asian westerly jet. Finally, the authors showed the different role of the Indian diabatic heating, depending on jet structures.

The authors used a large ensemble simulations to support the results revealed based on the observations. The large number of samples made the result more robust and reliable. The results are interesting and may help to improve understanding and prediction of sub-seasonal variability of East Asian summer rainfall. However, I have a major concern about the logic connection between the three parts in the Results section. First, I am a little confused about the connection between the results in sections 2.1 and 2.2. In section 2.1, the authors found a close relationship, on the interannual timescales, between the leading mode of East Asian rainfall in June and Indian rainfall. In section 2.2, the authors attempted to explain the interannual connection using the sub-seasonal variability of Indian diabatic heating, which is, however, not convincing. As described in Lines 151-152, for example, the dipole rainfall pattern in East Asia, which is the leading mode in June on the interannual timescales but is not prominent on the sub-seasonal timescales. Second, in section 2.3, the authors tried to analyze the interaction of CGT and wet band in East Asia using cluster analysis based on jet structure, and proposed the Indian diabatic heating anomaly can serve as a trigger or a reinforcer for wet band depending on if the warm advection is preexisting. What is the relationship between the jet structure and the preexisted warm advection? How does the jet structure affect different wave propagation illustrated in Figure 7, and how does the jet structure determine if the Indian diabatic heating is a trigger or a reinforcer for wet band in East Asia? The result of the different role of Indian diabatic heating, depending on jet structure, in wet band in East Asia is interesting, but the mechanisms need further clarification.

Minor comments

In section 2.2, the authors used the Q1 to represent the Indian summer monsoon heating. Why not use rainfall, which is more direct and maybe more accurate because the Q1 is calculated as the residual based on the thermodynamic equation?

What process is related to the dry band in East Asia since it is not correlated with the CGT wave pattern, Indian diabatic heating, and even rainfall in the western North Pacific (Fig. 6)? In addition, what is the relationship between the wet and dry band?

The authors showed that the dipole rainfall pattern in East Asia can exist without Indian diabatic heating (Fig. S6). So, how much can the Indian diabatic heating account for the dipole rainfall pattern variability?

Figure 6d: the title should be “V/T500”

L195: How does the anticyclonic anomaly cause the warm advection? Which component plays a dominant role, for example, the anomalous temperature advected by the climatological zonal/meridional winds or the climatological temperature advected by the anomalous zonal/meridional winds?

L206-208: what do you mean “the jet stream, which can also drive mechanically induced temperature advection that influences precipitation over East Asia”? Is the jet-drive mechanism different from the warm advection due to the anticyclone? If yes, what is the jet-drive mechanism?

Reviewer #1 (Remarks to the Author):

The authors attempt to find out whether the Indian summer monsoon (ISM) can serve as a predictor of the East Asian summer monsoon (EASM) on the subseasonal time scale in June. They argue that Indian subseasonal heating in June could enhance the East Asian rain band by triggering Rossby wave propagation along the subtropical jet and generating an anticyclone over the EA, which strengthens mid-tropospheric warm advection and lower-tropospheric moisture advection conducive to the EA rainband formation.

Response: We appreciate the time and effort of the reviewer who has put into careful reading and consideration of our manuscript, and we are grateful for the thoughtful comments and suggestions. The comments have significantly helped us to improve our manuscript and provide opportunity to see our research from a different perspective. The following is our point-to-point reply.

Overall assessment

The main conclusion of this work is flawed. The authors claimed that the Indian subseasonal heating in June is the critical factor controlling the dipole precipitation over EA. The authors presented Fig. 4 to support their claim. However, Figure 4 indicates the opposite: EA rainfall occurs before the ISM heating peak, ...

Response: As the reviewer pointed out, EA rainfall (wet band) can occur even before ISM heating maximum (lag0). This character is more evident in the HIST (Fig. S4 in the original manuscript, and Fig. S2 in the revised manuscript) than that in observation, as we have discussed in lines 245–249 in the original manuscript. This was presumably owing to the selected huge samples in HIST, totally 2,757 samples which includes cases that have EA wet band exist prior than lag0. Our K-means clustering classified some cases such as cluster 1 and 4 (Fig. 6 and Fig. 7 in the original manuscript; Fig. 3 and Fig. 4 in the revised manuscript) in which forcings other than ISM heating could induce EA wet band variability in the lead days (prior than lag0). These clusters showed that ISM heating can act as reinforcer of the rain band (line 218–222 in the original manuscript). We have revised the manuscript for better explanation of these findings (line 90–92 and line 250–254 in the revised manuscript).

... and when ISM heating decays, the EASM rainfall intensifies, meaning that ISM heating is not forcing EA rainfall.

Response: Thank you for your comments. After lag0 day (i.e., after the ISM heating maximum), the precipitation intensity is enhanced in all clusters (Fig. 6 in the original manuscript, or Fig. R1–2c for convenient). Therefore, the ISM heating is potentially to play a role in the modulation of EA wet band.

For better illustration the role of ISM heating on EA rainfall, we deepen the analysis to bridge them by considering CGT (circumglobal teleconnection). Here, a reference CGT pattern was created by applying the EOF analysis to seasonal mean (June–September) meridional wind at 200-hPa (V_{200}) over the domain of 20°–60°N, 0°–150°E, following the previous study (Yasui and Watanabe, 2010). We confirmed the EOF1 patterns from JRA-55 and HIST resemble each other and could represent CGT-like wave train; therefore, for convenience we used the HIST-based EOF1 pattern in the following figures (grey contours in Fig. R1–1; CGTI-PC1).

On lead 10 day, the CGT wave train propagates from Western Europe towards the ISM domain. At this stage, the East Asian CGT wave train is rather faint (Figs. R1–1a and b). On lag0 day, the vigorous ISM diabatic heating forces a wave train and dominates over East Asia (Figs. R1–1c and d), inducing a southerly wind that is prominent there. A few days later, the CGT emerges again but V200 anomalies are more intensive in the East Asian region, where the southerly wind still dominates. The southerly wind is important for EA rainfall, as we proposed in Fig. R1–2. The prominent southerly winds (Figs. R1–1 and R1–2e) promote the mid-tropospheric warm advection (color shading in Fig. R1–2d), as well as low-level moisture supply (Fig. R1–2f) to the EA, providing the preferential conditions for the wet band to form. We also confirmed these features of lag0 and lag10 are presented in each cluster but with slight difference in wavelength (figures omitted), in consistent with Fig. S5 in the original manuscript.

Furthermore, to confirm the role of ISM in the EA southerly winds as part of CGT, the linear baroclinic model (Watanabe & Kimoto, 2000; Watanabe & Jin, 2003; hereafter, LBM) experiments were carried out to see the atmospheric linear response to the ISM diabatic heating. We set the resolution of LBM to be T42L20 (i.e., 128×64 grids horizontally and 20 sigma levels vertically). The June basic state was used from the climatological values of JRA-55 from 1979 to 2020. The observed ISM diabatic heating of the June lag0 days averaged anomalies was imposed over the 10° – 30° N, 70° – 85° E region after regridding from JRA-55 to T42L20 resolution (Fig. R1–3). The LBM output well overlaps the wave train at lag10 (Fig. R1–1f, red contours), suggesting the wave train induced by diabatic heating of ISM is developed and trapped in the westerlies and propagates in phase with CGT. We trust these discussions, added to line 112–146 in the revised manuscript, can enrich the value of our findings.

Fig R1–1. The composite V200 anomalies on (a) lead10 day (b) lag0 day and (c) lag10 day for observation. (b), (d), and (f) are similar to (a), (c), and (e), but for HIST. Red contours in (f) are V200 anomalies from LBM experiment of June basic state, averaged from 30 to 50 days when the response has equilibrated. Grey contour shows the eigenvectors of CGTI–PC1. Dotted regions denote anomalies significant at the 90% and 99% confidence level for the observations and HIST, respectively.

Fig R1–2. HIST composite anomalies of (a) Q_1/C_p at lag0 days, (b) lag 3–4 days precipitation (shading) and 200-hPa geopotential height (contours), and temporal evolution from lead 10 day to lag 10 day of zonal mean (c) precipitation (shading) and 500-hPa temperature advection (contours, unit: K/day), (d) 500-hPa horizontal temperature advection (shading) and meridional temperature advection (contours, unit: K/day), (e) 500-hPa meridional wind (shading) and temperature (contours, unit: K), and (f) 850-hPa meridional wind (shading) and vertical integrated water vapor flux (arrows), for clusters 1–4 (columns from left to right, respectively). The zonal mean is the average over 105°–140°E. The stippled regions denote anomalies significant at the 99% confidence level based on the two-sided Student’s t test. Green hatching in (d) indicates vertical velocity (unit: Pa/s) lower than 0, significant at the 99% confidence level. “Nhist” indicates the number of samples used for the composite.

Fig R1-3. The diabatic forcing pattern that used in LBM.

In the revised manuscript, Figs. R1-1 and R1-2 was adopted as Figs. 2 and 4 in the main text, Fig. R1-3 was added to the revised supporting information as Fig. S10.

Reference:

- Watanabe, M., & Jin, F.-F. (2003). A Moist Linear Baroclinic Model: Coupled Dynamical–Convective Response to El Niño. *Journal of Climate*, 16(8), 1121–1139. [https://doi.org/10.1175/1520-0442\(2003\)16<1121:AMLBMC>2.0.CO;2](https://doi.org/10.1175/1520-0442(2003)16<1121:AMLBMC>2.0.CO;2)
- Watanabe, M., & Kimoto, M. (2000). Atmosphere-ocean thermal coupling in the North Atlantic: A positive feedback. *Quarterly Journal of the Royal Meteorological Society*, 126(570), 3343–3369. <https://doi.org/10.1256/smsqj.57016>
- Yasui, S., & Watanabe, M. (2010). Forcing processes of the summertime circumglobal teleconnection pattern in a dry AGCM. *Journal of Climate*, 23(8), 2093–2114. <https://doi.org/10.1175/2009JCLI3323.1>

The presentation is illogical. Section 2.1 (dominant mode of June East Asian precipitation and its controlling factor) presented three figures (Figs. 1, 2, and 3). They all show interannual variations of the June precipitation anomaly, associated water vapor fluxes, and diabatic heat sources. These results are well-known in many previous works and are nothing new. Besides, the three figures' results have nothing to do with the central theme (subseasonal variation). What caused the inter-

related variations was never discussed. Yet, the authors jumped to conclude, “The potential role of the ISM on the dipole precipitation pattern over East Asia is confirmed in the monthly mean field”. (Line 129-130).

Response: Thank you for your comments. In the previous version of the manuscript, we aimed to introduce a link between East Asian subseasonal rainfall variability and interannual variability as a theme to guide readers towards seasonal prediction. However, as suggested by the reviewer, we have decided to remove most of the interannual results in the revised manuscript as they are less relevant to the main objective of the study.

Figure R1–4 (now Figure 6 in the revised manuscript), confirm that stronger ISM forcing, indicated by higher ISMH values, correspond to higher PC1 scores for East Asian precipitation (EAPR–PC1) at interannual timescale. This is evidence denoting that subseasonal activity of ISM heating could lead to the modulation of interannual precipitation variability over East Asia. Description related to this has been added to discussion section (line 258–290), to underscore the potential significance of subseasonal ISM heating to interannual variation of East Asian rainband.

The subseasonal inter-related variations among CGT, ISM, and EASM have been discussed in the above replies (i.e., Figs. R1–1 to 1–3). We believe these can address the reviewer's concern.

Fig R1–4. Composite June monthly mean precipitation pattern based on observed (a) years with EAPR–PC1 greater than 0.5, (b) years with both EAPR–PC1 greater than 0.5 (EAPR–PC1+) and with detection of lag0 day, (c) years with EAPR–PC1 greater than 0.5 but without lag0 day, and (d) years with lag0 day only. (i) probability (right y-axis) of the interannual EAPR–PC1 (x-axis) distribution for all samples (black curve) and samples with lag0 (red curve), where the grey bars represent the EAPR–PC1 score (left y-axis) estimated from the conditioned lag0 ISMH magnitude for each 1K/day interval (x-axis). (e)–(h) and (j) are similar to (a)–(d) and (i), respectively, but for HIST. Dotted regions denote anomalies significant at the 90% and 99% confidence level for the observations and HIST, respectively. “N” indicates the number of samples used for the composite. The projected EAPR CGTI–PC1 values are shown on the top right of (d) and (h) based on Fig. S7 in the revised supporting information.

The citation of the literature in the introduction is ad hoc and does not acknowledge original contributions, reflecting an insufficient depth of the Authors' understanding and knowledge relevant to their study.

Response: We have included the following literatures in our revised introduction and acknowledged their contributions in the field related to our study. Listed below is ordered based on the sequence of appearance.

1. Wang, B. & Ho, L. Rainy season of the Asian-Pacific summer monsoon. *J. Clim.* **15**, 386–398 (2002).
5. Wang, B., Wu, R. & Fu, X. Pacific-East Asian teleconnection: How does ENSO affect East Asian climate? *J. Clim.* **13**, 1517–1536 (2000).
6. Wu, R., Hu, Z. Z. & Kirtman, B. P. Evolution of ENSO-related rainfall anomalies in East Asia. *J. Clim.* **16**, 3742–3758 (2003).
7. Yang, J., Liu, Q., Xie, S. P., Liu, Z. & Wu, L. Impact of the Indian Ocean SST basin mode on the Asian summer monsoon. *Geophys. Res. Lett.* **34**, 1–5 (2007).
14. Wang, B. *et al.* Fundamental challenge in simulation and prediction of summer monsoon rainfall. *Geophys. Res. Lett.* **32**, 2–5 (2005).
20. Zhang, R. Relations of Water Vapor Transport from Indian Monsoon with That over East Asia and the Summer Rainfall in China. *Adv. Atmos. Sci.* **18**, 1005–1017 (2001).
21. Kripalani, R. H. & Kulkarni, A. Monsoon rainfall variations and teleconnections over South and East Asia. *Int. J. Climatol.* **21**, 603–616 (2001).
23. Rodwell, M. J. & Hoskins, B. J. Monsoons and the dynamics of deserts. *Quart. J. R. Meteor. Soc.* **122**, 1385–1404 (1996).
24. Branstator, G. Circumglobal teleconnections, the jet stream waveguide, and the North Atlantic Oscillation. *J. Clim.* **15**, 1893–1910 (2002).
27. Mölg, T., Maussion, F. & Scherer, D. Mid-latitude westerlies as a driver of glacier variability in monsoonal High Asia. *Nat. Clim. Change* **4**, 68–73 (2014).
28. Krishnan, R. & Sugi, M. Baiu Rainfall Variability and Associated Monsoon Teleconnections. *J. Meteorol. Soc. Jp.* **79**, 851–860 (2001).
32. Kumar, K. K., Rajagopalan, B. & Cane, M. A. On the weakening relationship between the indian monsoon and ENSO. *Science* **284**, 2156–2159 (1999).

There are many other issues. However, the above concern makes me feel negative about the work. I recommend the rejection of the manuscript for publication.

Response: After making extensive revisions of the manuscript considering your valuable comments, we are confident that the quality of the study has significantly improved. We would like to kindly invite the reviewer to spend time to read our revised manuscript again.

Reviewer #2 (Remarks to the Author):

This is an interesting analysis with a primary emphasis on the role of the CGT in linking the ISM and ESAM on subseasonal time scales. According to their analysis and results, the authors suggested that diabatic heating anomalies associated with the ISM is critical to drive a regional component of the CGT over East Asia in June and synchronize rainfall variability along the path of the CGT from central India to East Asia. Actually, this physical connection and the role of the ISM in the formation of the CGT have been known for decades. The novelty of this new study primarily lies in its use of the K-mean clustering analysis and a large-ensemble simulation to reveal the connection of the two submonsoon systems over shorter time scales. However, a number of minor and major issues that are related to technical and scientific components of the study remain unclear and thus need to be addressed or clarified first before the manuscript can be considered for publication in the journal. Although the points I raised below regard many fundamental aspects of the formation of the CGT, I still feel that the study has strong potential to become an important piece of work that will greatly improve our physical understanding of the CGT.

Response: We appreciate the reviewer's valuable comments and encouragement. We have incorporated the reviewer's concerns and comments into the revised manuscript. Below is our point-by-point response.

1. The CGT is the leading NH circulation mode prevailing from June to September. The current study only focused on its variability in June. It is not clear to me how to apply what we learn from this month's situation to variations over other months.

Response: Thank you for your comments. Your comment motivated us to investigate the subseasonal ISM-CGT-EASM connection in other summer months. To achieve this, we analyzed the months of July to September by applying the same method with June. Here, we used standard deviation of daily ISMH values for each JRA-55 and HIST, considering different threshold value for each month as summarized in Table R2–1.

Table R2–1.

ISMH threshold (K/day) and sample size for each summer month for JRA-55 and HIST.

Month	1 std. dev. values (JRA-55 / HIST)	Threshold (JRA-55 / HIST)	Events (JRA-55 / HIST)
Jun	1.06 / 1.10	1.00 / 1.00	29 / 2757
Jul	0.98 / 0.79	1.00 / 0.80	25 / 1991
Aug	0.94 / 0.72	1.00 / 0.70	22 / 2200
Sep	0.99 / 0.77	1.00 / 0.80	26 / 2187

Lead–lag composite analyses were then performed using adjusted thresholds for each month (Figs. R2–1 and R2–2). Our results suggest that, in contrast to June, the ISM diabatic heating during July to September is weakly associated with the CGT circulation pattern at subseasonal

time scale. A meridional dipole-like precipitation pattern was found in the July observation, but not in the HIST. Meanwhile, the response of precipitation in August and September is generally much weaker than that in June.

Fig R2-1. Evolution of the observed composite of 200-hPa geopotential height (contours; units: m) and precipitation (color shading; units: mm/day) from June to September. N and Np indicates samples used for 200-hPa geopotential height and precipitation, respectively.

Fig R2-2. Similar to Figure R2-1, but for HIST.

To pursue whether the ISM could trigger East Asian CGT in other summer months, the linear baroclinic model (Watanabe & Kimoto, 2000; Watanabe & Jin, 2003; hereafter, LBM) experiments were carried out to see the atmospheric response to the ISM diabatic heating. The LBM is capable of simulating a linear response to a prescribed forcing, by removing nonlinearity of the processes. We set the resolution of LBM to be T42L20 (i.e., 128×64 grids horizontally and 20 sigma levels vertically). Horizontal diffusion with an e-folding time scale of 2 hours was applied.

We considered five scenarios (i.e., each month from May to September) with the 1979–2020 JRA-55 monthly mean climatological basic state. For each scenario, the same ISM diabatic heating of the June lag0 days averaged anomalies is imposed over the $10^\circ\text{--}30^\circ\text{N}$, $70^\circ\text{--}85^\circ\text{E}$ region, which is regridded from JRA-55 to T42L20 resolution (Fig R2–3).

The LBM output shows that in May and June, the ISM diabatic heating could trigger a regional CGT that propagates toward East Asia with “high-low-high” wave train pattern along the westerlies (Fig. R2–4). In contrast, the wave train does not exist in the LBM simulation for July to September. Furthermore, the East Asian anticyclone as a part of the CGT is shifted westwards from central-eastern China to the western edge of the Tibetan Plateau in simulations for summer months, which is no longer typical CGT patterns.

We have added Figs. R2–3 and R2–4 as Fig. S10 and S3, respectively, and the relevant descriptions to the revised manuscript (line 101–108 and line 390–399).

Fig R2–3. The diabatic heating pattern (units: K/day) that used in LBM.

Fig R2-4. The responses of the 200-hPa geopotential height (shading, units: m) and the wave activity flux (arrows) to the ISM diabatic heating simulated by LBM. Results are averaged from 30 to 50 days when LBM reached equilibrium. (a) May basic state, (b-e) similar to (a) but for June to September basic states were used, respectively.

Reference:

- Watanabe, M., & Kimoto, M. (2000). Atmosphere-ocean thermal coupling in the North Atlantic: A positive feedback. *Quarterly Journal of the Royal Meteorological Society*, 126(570), 3343–3369. <https://doi.org/10.1256/smsqj.57016>
- Watanabe, M., & Jin, F.-F. (2003). A Moist Linear Baroclinic Model: Coupled Dynamical–Convective Response to El Niño. *Journal of Climate*, 16(8), 1121–1139. [https://doi.org/10.1175/1520-0442\(2003\)16<1121:AMLBMC>2.0.CO;2](https://doi.org/10.1175/1520-0442(2003)16<1121:AMLBMC>2.0.CO;2)

2. In Fig1, the authors calculated the correlation of the blue and black curves and added some relevant discussion on this poor relationship. I don't fully understand why the authors took some effort to do this calculation since it is not expected to see a high correlation of these two time series given that these two curves are generated in very different climate scenarios.

Response: In the previous version of the manuscript, we aimed to introduce a link between East Asian subseasonal rainfall variability and interannual variability to guide readers towards seasonal prediction. In this revision, we have decided to remove most of the interannual results in the revised manuscript as they are less relevant to the main objective of the study.

Instead, an additional analysis was made in the revised manuscript presenting how subseasonal variability can affect interannual precipitation variability. Figure R2–5 (now Figure 6 in the revised manuscript) confirms that stronger ISM forcing, indicated by higher ISMH values, corresponds to higher PC1 scores for East Asian precipitation (EAPR–PC1) at interannual timescale. This is evidence denoting that subseasonal activity of ISM heating could lead to the modulation of interannual precipitation variability over East Asia. Description related to this has been added to the discussion section in the revised manuscript (line 258–290).

Fig R2–5. Composite June monthly mean precipitation pattern based on observed (a) years with EAPR–PC1 greater than 0.5, (b) years with both EAPR–PC1 greater than 0.5 (EAPR–PC1+) and with detection of lag0 day, (c) years with EAPR–PC1 greater than 0.5 but without lag0 day, and (d) years with lag0 day only. (i) probability (right y-axis) of the interannual EAPR–PC1 (x-axis) distribution for all samples (black curve) and samples with lag0 (red curve), where the grey bars represent the EAPR–PC1 score (left y-axis) estimated from the conditioned lag0 ISMH magnitude for each 1K/day interval (x-axis). (e)–(h) and (j) are similar to (a)–(d) and (i), respectively, but for HIST. Dotted regions denote anomalies significant at the 90% and 99% confidence level for the observations and HIST, respectively. “N” indicates the number of samples used for the composite. The projected EAPR–PC1 values are shown on the top right of (d) and (h) based on Fig. S7 in the revised supporting information.

3. It is still not clear to me what we can learn from the clustering analysis. If I understand the method right, these four modes are not orthogonal to each other and there are some common features shared by these four modes. If so (they look very similar in Fig. 6 & 7), I feel the author should put more attention on these common features and also distinctions of each mode from the common features.

Response: Thank you for your interesting comments. Our reply is twofold as below.

i. Are clustered jet features orthogonal?

We have created 4 clusters by inputting the values of latitude, longitude, and corresponding horizontal wind speed at 200-hPa along the jet axis into the K-means clustering. As the reviewer pointed out, the K-means clustering does not necessarily ensure that the clustered patterns are orthogonal, as it is a distance-based algorithm. To test the orthogonality of the clustered result, the same jet axes data were input to the PCA analysis. The first two leading modes explained 20.95% and 10.43% of the total variance, respectively. By comparing the result of the K-means clustering and two leading modes of the PCA analysis, we found a clear relationship between these two methods (Fig. R2–6). The group averages for clusters 1 and 3 correspond well to the negative PC2 and the positive PC2, while the group averages of clusters 2 and 4 are similarly related to the negative and positive PC1. Therefore, mean characteristics of four clusters can be assumed to be orthogonal, although the individual samples may not be.

We have added Fig. R2–6 as Fig. S12 in the supporting information and relevant description to Line 414–416 in the revised manuscript.

Fig R2–6. Comparison of four clusters about the jet pattern created by K-means clustering and two leading modes from PCA analysis. The percentage values mean the proportion of explanation for PC1 and PC2. The crosses represent the group mean of each cluster.

ii. What can we learn from the clustering analysis?

Our claim is that the response of downstream atmospheric circulation over East Asia to upstream ISM heating is different according to the pathway of the jet stream. The vigorous ISM diabatic heating at lag0 perturbs the air close to the westerly jet in June and plays a role as Rossby wave source. Since westerly jet stream can act as a waveguide (Hoskins and Ambrizzi, 1993), it

could confine the wave and direct it toward East Asia. Our results depict that clusters 2 and 3 show southward bending of the jet over East Asia, hence, the wave train is heading to the south, forming a cold anomaly (negative geopotential height anomaly) center over East Asian sector on lag0 (Figs. R2–7b and c). The cold center is consequently positioned southward of the jet (south of 35°N). In contrast, clusters 1 and 4 represent the jet running straight, and thus, the wave tends to propagate zonally toward East Asia. This condition leads to the position of the cold center to be more north over the East Asian sector (Figs. R2–7a and d). Such difference in the position of cold center at 500h-hPa relates to the behavior of the rainband as pointed out in our original manuscript since it affects the warm advection as discussed below.

In the revised manuscript, we have deepened the analysis on warm advection at 500-hPa. The analysis was performed to elucidate which wind or temperature is attributable to intensify the advection. Here we use climatological fields of wind and temperature to test how each of them can modulate the temperature advection. The result (Fig. R2–8d) reveals that the anomalous wind is the main factor regulating the anomalous temperature advection in all clusters. The sign of the total temperature advection is, however, dependent on the magnitude of the cold advection induced by anomalous temperature. This temperature-induced cold advection is found to be controlled by the position and intensity of the cold center as part of the wave train in East Asia (Fig. R2–8 b and e). This cold center exists in the lead days for all clusters but with different latitudinal position. In clusters 2 and 3 the cold center is located close to the wet band (35°N), while it is located north of the wet band in clusters 1 and 4 (Fig. R2–8b and e). These positioning of cold center nicely correspond to the pathway of the wave train, and thus reflect the jet structure in each cluster. Through above-mentioned mechanism, the ISM diabatic heating could modulate the East Asian rain band in a different manner in accordance with the jet structure.

Fig. R2–7 was added to the supporting information as Fig. S5. We have updated Fig. R2–8 as Fig. 4 and the relevant descriptions to line 183–194 and line 215–218 in the revised manuscript. We trust these discussions can enrich the value of our findings.

Fig R2–7. HIST composite anomalies of event0 jet axis (thick red line), lag0 200-hPa geopotential height anomalies (shading), and lag0 200-hPa meridional wind anomalies (contours) together with wave activity flux of lag0 days (arrows). (a–d) are for clusters 1–4. (e) and (f) depicts lag0 days’ zonal and meridional component of the wave activity flux, respectively, for each cluster averaged over 30–40°N.

Fig R2-8. HIST composite anomalies of (a) Q_1/C_p at lag0 days, (b) lag 3–4 days precipitation (shading) and 200-hPa geopotential height (contours), and temporal evolution from lead 10 day to lag 10 day of zonal mean (c) precipitation (shading) and 500-hPa temperature advection (contours, unit: K/day), (d) 500-hPa temperature advection with climatological wind field (shading) and climatological temperature field (contours, unit: K/day), (e) 500-hPa meridional wind (shading) and temperature (contours, unit: K), and (f) 850-hPa meridional wind (shading) and vertical integrated water vapor flux (arrows), for clusters 1–4 (columns from left to right, respectively). The zonal mean is the average over 105°–140°E. The stippled regions denote anomalies significant at the 99% confidence level based on the two-sided Student’s t test. Green hatching in (d) indicates vertical velocity (unit: Pa/s) lower than 0, significant at the 99% confidence level. “Nhist” indicates the number of samples used for the composite.

Reference:

Hoskins, B. J., & Ambrizzi, T. (1993). Rossby Wave Propagation on a Realistic Longitudinally Varying Flow. *Journal of the Atmospheric Sciences*, 50(12), 1661–1671. [https://doi.org/10.1175/1520-0469\(1993\)050<1661:RWPOAR>2.0.CO;2](https://doi.org/10.1175/1520-0469(1993)050<1661:RWPOAR>2.0.CO;2)

4. The study emphasized that the CGT mode presented here varies energetically on subseasonal time scales, using a lead-lag composite method covering data from day-10 to day+10. In the paper, it is better to further illustrate what the main periodicity of this feature is. A power spectrum analysis of the time series related to this subseasonal variability may help the authors answer this question.

Response: Thank you for your interesting comments. We are very much interested in this issue. Hence, we conducted additional analysis on the periodicity of CGT. The procedures and results are explained as below.

i. Creating a CGT index

We first create a CGT reference pattern by applying EOF analysis to seasonal mean (June–September) meridional wind at 200-hPa (V200) over the domain of 20°–60°N, 0°–150°E, following the previous study (Yasui and Watanabe, 2010). The EOF1 patterns from JRA-55 and HIST resemble each other (Fig. R2–9). Hence, we used HIST EOF1 pattern in the following analysis for convenience. We then project the daily June V200 anomalies onto this reference pattern to obtain the daily CGT index (hereafter, CGTI–PC1) for both JRA-55 and HIST. The CGTI–PC1 represents the subseasonal variation of CGT wave pattern. We also compute another CGT index for comparison, which was introduced by Ding and Wang, (2005), by area-averaging the 200-hPa geopotential height over 35°–40°N, 60°–70°E (hereafter, CGTI–DW).

Fig R2–9. The EOF1 eigenvectors of interannual June to September seasonal mean meridional wind anomalies at 200-hPa. The contour is the result for JRA-55 (interval: 0.01) and the color shading is for the result of HIST, suggesting CGT patterns are similar between JRA-55 and HIST.

ii. Subseasonal variation of CGT indices

In order to track the variation of CGT associated with ISM activities, Fig. R2–10 shows mean CGTI variations from a lead of 20 day to a lag of 20 day. For both CGTI–PC1 and CGTI–DW, the composite mean variations show an "M" shape (i.e., peak-valley-peak), meaning strengthening CGT activities in pre- and post-lag0 days. Power spectrum analysis for these CGTIs shows a power around 0.06–0.08 (day⁻¹) which roughly corresponds to 10–20 day (Fig. R2–10c and d). Furthermore, the power spectrum for each event show there is a significant power (i.e., exceeding 95% red noise significance boundary) in the 5–25-day range (Fig. R2–10e), with a dominant mode of 0.07 day⁻¹ (Fig. R2–10f). These results suggest that the CGT behaves as a

quasi-biweekly oscillation.

Fig R2–10. Time variation of circum-global teleconnection from lead20 to lag20 days represented by (a) CGTI–PC1 and (b) CGTI–DW. (c) to (d) power spectrum for observation and HIST, respectively, where the dash-dot and dashed red curve denote the red noise and its 95% significance boundary. (e) A list of variance values for each event. The range of significant frequency (day^{-1}) exceeding the 95% red noise confidence interval from the power spectrum are plotted, and the color shading represent variance. The black dashed vertical line is for the demarcation of the events collected from observation (left) and HIST (right). (f) probability distribution of the frequency based on (e).

iii. Mechanism of subseasonal variation of CGT

Finally, the possible mechanism of the subseasonal CGT variation is discussed. The first peak of the "M" shape for the subseasonal CGT variation denotes the propagation of the CGT wave train from western Europe toward the ISM domain. At this stage, the East Asian CGT wave train is still weak (Fig. R2–11). On lag0 day, the vigorous ISM diabatic heating forces a wave train which becomes more pronounced over East Asia than in prior days. The wave train, however, is orthogonal (e.g., half-phase shifted) to CGT at lag0, as denoted by the valley of the "M" shape (Fig. R2–10a and b). Few days later, the CGT emerges again but much more energetically in the East Asian region. Furthermore, the LBM output for June basic state well overlaps the wave train at lag10 day, suggesting the wave train induced by diabatic heating of ISM is developed and trapped in the westerlies, propagates in phase with CGT.

Overall, our additional analysis suggests that CGT wave train interacting with ISM diabatic heating exhibits a character of quasi-biweekly oscillations. The first phase is the propagation of wave train from western Europe, far upstream of the ISM domain. The second phase is the active of the ISM convection and diabatic heating which excites the wave train and maintains propagation to East Asia. Here, ISMH activities might be related to European-CGT wave train as

illustrated by Ding and Wang, (2007, 2009). Finally, the ISM-induced wave train takes on a CGT-like pattern as it extends horizontally along the westerly jet. Very interestingly, the result at lag10 well overlaps the wave train simulated by LBM using June basic state (Fig. R2–11f). The transition of the wave pattern between lag0 and lag10 over East Asia is proposed to be linked to ISMH activities as similar transition pattern was seen in the LBM simulation (results omit). In the revised manuscript, we have added Fig. R2–11 as Fig. 2, and relevant description to Line 112–146. Figs. R2–9 and R2–10 is added as Figs. S8 and S4, respectively, to the supporting information.

Fig R2–11. The composite V200 anomalies on (a) lead10 day (b) lag0 day and (c) lag10 day for observation. (b), (d), and (f) are similar to (a), (c), and (e), but for HIST. Red contours in (f) are V200 anomalies from LBM experiment of June basic state, averaged from 30 to 50 days when model integration reached equilibrium. Grey contour shows the eigenvectors of CGTI-PC1 (Fig. R2–9). Dotted regions denote anomalies significant at the 90% and 99% confidence level for the observations and HIST, respectively.

Reference:

Ding, Q., & Wang, B. (2005). Circumglobal teleconnection in the Northern Hemisphere summer. *Journal of Climate*, 18(17), 3483–3505. <https://doi.org/10.1175/JCLI3473.1>

Ding, Q., & Wang, B. (2007). Intraseasonal teleconnection between the summer Eurasian wave train and the Indian Monsoon. *Journal of Climate*, 20(15), 3751–3767. <https://doi.org/10.1175/JCLI4221.1>

Ding, Q., & Wang, B. (2009). Predicting extreme phases of the Indian summer monsoon. *Journal of Climate*, 22(2), 346–363. <https://doi.org/10.1175/2008JCLI2449.1>

Yasui, S., & Watanabe, M. (2010). Forcing processes of the summertime circumglobal teleconnection pattern in a dry AGCM. *Journal of Climate*, 23(8), 2093–2114. <https://doi.org/10.1175/2009JCLI3323.1>

5. Some minor wording issues pop up here or there in the current form. I prefer to place my main focus on these wording problems next round after these scientific issues I just raised could be ironed out.

Response: We made a thorough self-check of the text. We trust the readability has been improved and wording becomes more appropriate.

Reviewer #3 (Remarks to the Author):

Comment on “East Asian summer rainfall stimulated by subseasonal Indian monsoonal heating” by Li et al.

The authors found that on the sub-seasonal timescales, an Indian anomalous diabatic heating may serve as a predictor for East Asian rainfall in early summer. The authors first identified the leading mode of June rainfall in East Asia on the interannual timescales and revealed the relation to Indian rainfall. Then, the authors proposed that the Indian diabatic heating anomaly can lead to East Asian rainfall variability through a wave train along the Asian westerly jet. Finally, the authors showed the different role of the Indian diabatic heating, depending on jet structures. The authors used a large ensemble simulation to support the results revealed based on the observations. The large number of samples made the result more robust and reliable. The results are interesting and may help to improve understanding and prediction of sub-seasonal variability of East Asian summer rainfall. However, I have a major concern about the logic connection between the three parts in the Results section.

Response: We thank the reviewer for the positive evaluation of our study, and we also appreciate the reviewer’s comments that help us improve the manuscript quality. We have revised the manuscript carefully according to the comments. The following is our point-by-point response.

First, I am a little confused about the connection between the results in sections 2.1 and 2.2. In section 2.1, the authors found a close relationship, on the interannual timescales, between the leading mode of East Asian rainfall in June and Indian rainfall. In section 2.2, the authors attempted to explain the interannual connection using the sub-seasonal variability of Indian diabatic heating, which is, however, not convincing. As described in Lines 151-152, for example, the dipole rainfall pattern in East Asia, which is the leading mode in June on the interannual timescales but is not prominent on the sub-seasonal timescales.

Response: In the previous version of the manuscript, we aimed to introduce a link between East Asian subseasonal rainfall variability and interannual variability to guide readers towards seasonal prediction. However, as suggested by the reviewer, we have decided to remove most of the interannual results in the revised manuscript as they are less relevant to the main objective of the study.

As Fig. R3–1 (now Figure 6 in the revised manuscript) shows, stronger ISM forcing indicated by higher ISMH values corresponds to higher PC1 scores for East Asian precipitation (EAPR–PC1) at interannual timescale. This is nice evidence denoting that subseasonal activity of ISM heating could lead to modulation of interannual precipitation variability over East Asia. Description related to this figure has been added to discussion section (line 271–290) to underscore the potential significance of subseasonal ISM heating to interannual variation of East Asian rainband.

Fig R3–1. Composite June monthly mean precipitation pattern based on observed (a) years with EAPR–PC1 greater than 0.5, (b) years with both EAPR–PC1 greater than 0.5 (EAPR–PC1+) and with detection of lag0 day, (c) years with EAPR–PC1 greater than 0.5 but without lag0 day, and (d) years with lag0 day only. (i) probability (right y-axis) of the interannual EAPR–PC1 (x-axis) distribution for all samples (black curve) and samples with lag0 (red curve), where the grey bars represent the EAPR–PC1 score (left y-axis) estimated from the conditioned lag0 ISMH magnitude for each 1K/day interval (x-axis). (e)–(h) and (j) are similar to (a)–(d) and (i), respectively, but for HIST. Dotted regions denote anomalies significant at the 90% and 99% confidence level for the observations and HIST, respectively. “N” indicates the number of samples used for the composite. The projected EAPR–PC1 values are shown on the top right of (d) and (h) based on Fig. S7 in the revised supporting information.

Second, in section 2.3, the authors tried to analyze the interaction of CGT and wet band in East Asia using cluster analysis based on jet structure, and proposed the Indian diabatic heating anomaly can serve as a trigger or a reinforcer for wet band depending on if the warm advection is preexisting. What is the relationship between the jet structure and the preexisted warm advection?

Response: Thank you for your constructive comments. We first invite the reviewer to our revised manuscript for some updated results. To be concise, we claim that the ISM diabatic heating has an effect to maintain the CGT wave train that propagates toward East Asia. This feature is confirmed by two additional analyses: composite analysis and linear baroclinic model experiment (Figure 2 and Line 125–136 in the revised manuscript). As we propose in the revised manuscript, southerly wind over East Asia induced by ISM heating is important because it promotes mid-tropospheric warm advection near the wet band and fuels the low-level moisture to the wet band.

The leading pattern for summer East Asian jet variability is known as the meridional displacement of the Asian jet (JMD) (e.g., Hong and Lu, 2016; Chowdary et al., 2021). In the clusters, we have observed that the East Asian jets exhibit JMD-related characteristics. Specifically, in clusters 1 and 4, the north JMD is observed as the East Asian jets tend to run further north, whereas in clusters 2 and 3, the south JMD is observed as the East Asian jets tend to travel further south (Fig. R3–2). The north and south JMD are also related to meridional wind anomalies; specifically, the former corresponds to tropospheric southerly winds prevailing in East Asia (i.e., Figs. R3–4e and f in lead days), which is associated to warm advection together with enhanced moisture supply near the wet band, while the latter does not show such feature (Fig. R3–2 bottom). These results are in consistent with previous studies about East Asian JMD (e.g., Wang et al., 2018, Wang et al., 2019). In clusters 1 and 4, such beneficial conditions allow for the formation of the wet band in lead days, and it pre-exists. In contrast, for clusters 2 and 3, the wet band has not yet formed during the lead days because the southerly wind is not dominant in East Asia.

We have incorporated above discussions as text near line 161–164 and 209–211 in the revised manuscript.

Fig R3–2. Event0 composite of 200-hPa reconstructed horizontal wind (top row), anomalies of precipitation and 200-hPa horizontal wind (middle row), and 500-hPa thermal advection with water vapor flux of clusters from 1 to 4 (columns from left to right, respectively). The red curves stand for the jet axis for each cluster.

Reference:

- Chowdary, J. S., Vibhute, A. S., Darshana, P., Parekh, A., Gnanaseelan, C., & Attada, R. (2021). Meridional displacement of the Asian jet and its impact on Indian summer monsoon rainfall in observations and CFSv2 hindcast. *Climate Dynamics*, 58(3–4), 811–829. <https://doi.org/10.1007/s00382-021-05935-1>
- Hong, X., & Lu, R. (2016). The meridional displacement of the summer Asian jet, Silk Road pattern, and tropical SST anomalies. *Journal of Climate*, 29(10), 3753–3766. <https://doi.org/10.1175/JCLI-D-15-0541.1>
- Wang, S., Zuo, H., Zhao, S., Zhang, J., & Lu, S. (2018). How East Asian westerly jet's meridional position affects the summer rainfall in Yangtze-Huaihe River Valley? *Climate Dynamics*, 51(11–12), 4109–4121. <https://doi.org/10.1007/s00382-017-3591-3>
- Wang, S., Zuo, H., Yin, Y., Wang, J., & Ma, X. (2019). Asymmetric impact of East Asian jet's variation on midsummer rainfall in North China and Yangtze River Valley. *Climate Dynamics*, 53(9–10), 6199–6213. <https://doi.org/10.1007/s00382-019-04923-w>

How does the jet structure affect different wave propagation illustrated in Figure 7, and how does the jet structure determine if the Indian diabatic heating is a trigger or a reinforcer for wet band in East Asia? The result of the different role of Indian diabatic heating, depending on jet structure, in wet band in East Asia is interesting, but the mechanisms need further clarification.

Response: Thank you for your constructive comments. Our reply for each of your question is arranged below.

i. How does the jet structure affect different wave propagation?

As we proposed in the previous comment, the major difference among clusters is the JMD. Clusters 1 and 4 are related north JMD (Figs. R3–2a and d), while the south JMD is found in clusters 2 and 3 over the East Asian domain (Figs. R3–2b and c).

Our claim is that the response of downstream atmospheric circulation over East Asia to upstream ISM heating is different according to the pathway of the jet stream. The vigorous ISM diabatic heating at lag0 perturbs the air close to the westerly jet in June and plays a role as Rossby wave source. Since westerly jet stream can act as a waveguide (Hoskins and Ambrizzi, 1993), it could confine the wave and direct it toward East Asia. Our results depict that clusters 2 and 3 show southward bent of the jet over East Asia (i.e., south JMD), hence, the wave train is heading to the south, forming a cold anomaly (negative geopotential height anomaly) center over East Asian sector on lag0 (Fig. R3–3 b and c). The cold center is consequently positioned southward of the jet (south of 35°N). In contrast, clusters 1 and 4 represent the jet running straight (namely a north JMD scenario), and thus, the wave tends to propagate zonally toward East Asia. This condition leads to the position of the cold center to be more north over the East Asian sector (Figs. R3–3a and d). Such difference in the position of cold center at 500h-hPa relates to the behavior of the rainband since it affects the warm advection as discussed in the next (please see our reply in the section ii in the next).

In the revised manuscript, Fig. R3–3 was adopted as Fig. S5, and relevant description were

added in line 184–194 in the revised manuscript.

Fig R3–3. HIST composite anomalies of event0 jet axis (thick red line), lag0 200-hPa geopotential height anomalies (shading), and lag0 200-hPa meridional wind anomalies (contours) together with wave activity flux of lag0 days (arrows). (a–d) are for clusters 1–4. (e) and (f) depicts lag0 days’ zonal and meridional component of the wave activity flux, respectively, for each cluster averaged over 30–40°N.

ii. how does the jet structure determine if the Indian diabatic heating is a trigger or a reinforcer for wet band in East Asia?

As above, the jet structure determines the position of cold anomalies over East Asia. Here, we explain how it potentially relates to trigger/reinforcer role of ISH. To achieve that, we have deepened the analysis on warm advection at 500-hPa in the revised manuscript. We use climatological fields of wind and temperature to test how each of them can contribute the temperature advection. The result (Fig. R3–4d) reveals that the anomalous wind is the main factor regulating the anomalous temperature advection in all clusters. The sign of the advection is, however, dependent on the location and magnitude of the cold advection induced by anomalous temperature. Namely, this temperature-induced cold advection is found to be controlled by the position and intensity of the cold center in East Asia as part of the wave train (Figs. R3–4 b and e). This cold center exists in the lead days for all clusters but with different latitudinal position. In clusters 2 and 3 the cold center is located close to the wet band (35°N), while it is located north of the wet band in clusters 1 and 4 (Figs. R3–4b and e). These positionings of cold center nicely correspond to the pathway of the wave train, and thus reflect the role of jet structure in each cluster. Through above-mentioned mechanism, the ISM diabatic heating could modulate the East Asian rainband in a different manner in accordance with the jet structure.

In conclusion, we found that the jet pattern in the preconditions (i.e., event0) is important. It first determines the large-scale temperature advection in the lead days. Moreover, the jet structure guides the ISM-induced CGT wave train and determines the location of cold center over East Asian sector. Under south JMD condition (i.e., southward bending jet; clusters 2 and 3), the cold anomaly hinders the prevalence of warm advection in East Asia, could result in the ISM acting as the trigger. Our clustering results (together sections i and ii) suggest jet structure is important in

subseasonal prediction of ISM-EASM connection.

We have added Fig. R3–4 as Fig. 4, and the relevant descriptions to the revised manuscript (line 215–218).

Fig R3–4. HIST composite anomalies of (a) Q_1/C_p at lag0 days, (b) lag 3–4 days precipitation (shading) and 200-hPa geopotential height (contours), and temporal evolution from lead 10 day to lag 10 day of zonal mean (c) precipitation (shading) and 500-hPa temperature advection (contours, unit: K/day), (d) 500-hPa horizontal temperature advection (shading) and meridional temperature advection (contours, unit: K/day), (e) 500-hPa meridional wind (shading) and temperature (contours, unit: K), and (f) 850-hPa meridional wind (shading) and vertical integrated water vapor flux (arrows), for clusters 1–4 (columns from left to right, respectively). The zonal mean is the average over 105°–140°E. The stippled regions denote anomalies significant at the 99% confidence level based on the two-sided Student’s t test. Green hatching in (d) indicates vertical velocity (unit: Pa/s) lower than 0, significant at the 99% confidence level. “Nhist” indicates the number of samples used for the composite.

Reference:

Hoskins, B. J., & Ambrizzi, T. (1993). Rossby Wave Propagation on a Realistic Longitudinally Varying Flow. *Journal of the Atmospheric Sciences*, 50(12), 1661–1671. [https://doi.org/10.1175/1520-0469\(1993\)050<1661:RWPOAR>2.0.CO;2](https://doi.org/10.1175/1520-0469(1993)050<1661:RWPOAR>2.0.CO;2)

Minor comments

In section 2.2, the authors used the Q1 to represent the Indian summer monsoon heating. Why not use rainfall, which is more direct and maybe more accurate because the Q1 is calculated as the residual based on the thermodynamic equation?

Response: Thank you for your comments. We carefully compared the relationship between precipitation anomaly index (PRI) and ISMH. The correlation coefficients between June daily Indian PRI (area-averaged over 10°–30°N, 70°–85°E) and daily ISMH (same domain as PRI but with mass-weighted integration from 850 to 200 hPa) are 0.80 and 0.99 (both with $p < 0.01$), respectively, for APHRODITE PRI versus JRA-55 ISMH and HIST PRI versus HIST ISMH. This result indicates ISMH is a good proxy of precipitation in ISM domain. The ISMH we defined is a diagnosis capable to represent the gradual evolution of the ISMH corresponding to large-scale circulation. Furthermore, the ISMH is a substitute for precipitation index when in situ observation is lacking (e.g., over oceans or stations are sparse), in particular considering the analysis period of this study spanning from 1958-2020. We have added a brief description to line 381–388 (in method section).

What process is related to the dry band in East Asia since it is not correlated with the CGT wave pattern, Indian diabatic heating, and even rainfall in the western North Pacific (Fig. 6)? In addition, what is the relationship between the wet and dry band?

Response: We are also interested in this question. The relationship among wet band, dry band, and ISMH are illustrated in Fig. R3–5, by computing lead-lag correlation of dry band PRI with respective to wet band PRI and ISMH. The results (both observation and HIST) indicate the dry band and wet band are likely to occur simultaneously, we therefore speculate that the development of wet band and dry band might reflect a northward location/migration of the rand band.

Figure R3–5 also denotes that the dry band tends to be more prominent before the ISM heating becomes vigorous. This may be related to the activities of low-level easterly before the ISM to be active (Fig. R3–4f; lower left corner for each panel). We have added this argument to line 306–307 in the revised manuscript.

Fig R3–5. June subseasonal lead-lag correlations between wet band precipitation anomaly index and ISMH, respectively, relative to dry band precipitation anomaly index based on (a) observation data (APHRODITE) and (b) HIST data. SPr and NPr are for precipitation anomalies averaged over dry band and wet band regions, respectively. SPr lead-lag NPr means lead-lag correlations of SPr with respect to NPr, similarly for SPr lead-lag ISMH. Values with solid circle satisfy $p < 0.01$.

The authors showed that the dipole rainfall pattern in East Asia can exist without Indian diabatic heating (Fig. S6). So, how much can the Indian diabatic heating account for the dipole rainfall pattern variability?

Response: We attempted to estimate this ratio based on HIST data to sample more cases. The procedures are as followings. We first calculated the daily-basis EAPR-PC1 score as a projection of June subseasonal precipitation anomalies onto the interannual June EOF1 dipole pattern mode (Fig. S9 in the revised supporting information). Then, we counted the number of days after lag0 with projected $PC1 > 0.5$ (21,852 days). Among those, days with high PC1 are assumed relevant to the ISM diabatic heating since they occurred after the lag0. Meanwhile, the total days with $PC1 > 0.5$ are 77,494 days. Therefore, this ratio is about 28.20%. In conclusion, ISM heating may link to more than one fourth of the subseasonal dipole events in June. While we have made our best attempt, it is still a rough estimation. We therefore decide not to add this result to the main text; instead, we added few words (line 280–281 in revised manuscript), based on the EAPR-PC1 score of lag0 composite (Fig. R3–1h), to emphasize the possibility that the interannual variability of the dipole pattern can be explained by the subseasonal ISM heating.

Figure 6d: the title should be “V/T500”

Response: we have corrected the title.

L195: How does the anticyclonic anomaly cause the warm advection? Which component plays a dominant role, for example, the anomalous temperature advected by the climatological zonal/meridional winds or the climatological temperature advected by the anomalous zonal/meridional winds?

Response: Thank you for your constructive comments., and we believe we have addressed the temperature advection related questions in our reply to the reviewer’s major comment#2 (Fig. R3–4f).

L206-208: what do you mean “the jet stream, which can also drive mechanically induced temperature advection that influences precipitation over East Asia”? Is the jet-drive mechanism different from the warm advection due to the anticyclone? If yes, what is the jet-drive mechanism?

Response: We have deleted this sentence to avoid confusion. We believe we have addressed this question in reviewer’s major concern#1 and #2 (Figs. R3–2, R3–3, and R3–4d).

REVIEWER COMMENTS

Reviewer #2 (Remarks to the Author):

The paper has been improved this round with all issues i brought up well addressed. Thus I recommend publication of the manuscript.

Reviewer #3 (Remarks to the Author):

The authors have revised the manuscript thoroughly and focused on connection between ISM heating and East Asian early-summer precipitation through the CGT pattern on the sub-seasonal timescales. They analyzed feature and evolution of the sub-seasonal CGT pattern and highlighted the associated southerly winds and warm advection over East Asia play a dominant role in modulating sub-seasonal variability of precipitation in East Asia. They further show that the ISM heating can act as a reinforcer or trigger of East Asian precipitation depending on the different jet structures. The results have greatly advanced our understanding of sub-seasonal variability of mid-latitude circulation and Indian-East Asian summer monsoons connection. However, I still have doubts on the conclusions, including the dominant effect of southerly wind over East Asia and the underlying mechanisms on the different roles of ISM heating depending on the jet structures. Therefore, a major revision is recommended.

Major comments

1. Similarity in the total temperature advection anomalies (Fig. 4c) and that by climatological wind (shading Fig. 4d) indicates that the temperature anomaly, instead of wind anomaly, is the major factor for the warm advection and precipitation in East Asia. So, the result is different with the conclusion in the abstract (Lines 24-25). Moreover, the precipitation anomaly distributes along a zonal-oriented band rather than along a meridional-oriented band, which suggests that the warm advection by the zonal wind is likely stronger than that by meridional wind. I agree with the moisture transport role of the southerly wind, but doubt the dominant role of the southerly wind in the warm advection at 500 hPa over East Asia. I suggest that the authors present the spatial pattern and quantitative result of temperature advection by different components.

2. This study has shown the effect of ISM heating on early-summer precipitation in East Asia, on the sub-seasonal timescales. The results are important for understanding the sub-seasonal variability of the East Asian summer precipitation. I wonder if the authors can quantify the effect of the ISM heating. I am curious about the two aspects: (1) probability for wet events in the north wet band due to the ISM heating and (2) ratio of intraseasonal precipitation in the north wet band can be explained by the ISM heating. The quantitative results may provide a straightforward view of the ISM heating on sub-seasonal precipitation variability in East Asia.

3. One major conclusion of this study is that the jet structures modulate the role of ISM heating as a trigger or reinforcer of the rainband. The authors do show different preceding signals in the mid-high latitude Eurasia for four different jet structures (Fig. 5). Though the signals are clear, I wonder the underlying mechanisms. I understand the different locations of jet axis and different jet intensity may affect the latitudinal and longitudinal locations of the CGT portion over East Asia. But how can the jet structures in clusters 1 and 4 lead to preexisting southerly winds and warm advection, but that in clusters 2 and 3 cannot?

Minor comments:

1. Line 24: Why is the northward tilt of southerly wind over East Asia so important to be presented in the abstract? In addition, please explain the possible reason.

2. Fig. 5: U200 anomalies are absent.

3. The authors used two composite analysis based on event0 and lag0 day, in which the former is only used for the jet cluster analysis. Why were the two different composite methods used? The different composite result cannot be explained directly for each other. If possible, please use one for all the analysis in one paper. Otherwise, the reason and the possible effect using different

composites on results should be discussed.

Reviewer #3 (Remarks to the Author):

The authors have revised the manuscript thoroughly and focused on connection between ISM heating and East Asian early-summer precipitation through the CGT pattern on the sub-seasonal timescales. They analyzed feature and evolution of the sub-seasonal CGT pattern and highlighted the associated southerly winds and warm advection over East Asia play a dominant role in modulating sub-seasonal variability of precipitation in East Asia. They further show that the ISM heating can act as a reinforcer or trigger of East Asian precipitation depending on the different jet structures. The results have greatly advanced our understanding of sub-seasonal variability of mid-latitude circulation and Indian-East Asian summer monsoons connection. However, I still have doubts on the conclusions, including the dominant effect of southerly wind over East Asia and the underlying mechanisms on the different roles of ISM heating depending on the jet structures. Therefore, a major revision is recommended.

Response: We sincerely appreciate the time and effort the reviewer has dedicated to thoroughly reading and considering our manuscript. We are grateful for the valuable comments and words of encouragement. In this revision, we have presented the spatial pattern of temperature advection anomalies, the relationship between jet meridional displacement and meridional wind, and quantitative result of the potential impact of ISM on EASM precipitation. The reviewer's concerns and comments have been incorporated into the revised manuscript. Below is our point-by-point response.

Major comments

1. Similarity in the total temperature advection anomalies (Fig. 4c) and that by climatological wind (shading Fig. 4d) indicates that the temperature anomaly, instead of wind anomaly, is the major factor for the warm advection and precipitation in East Asia. So, the result is different with the conclusion in the abstract (Lines 24-25).

Response: We are grateful to the reviewer for bringing this issue to our attention. Our error in the caption of Fig. 4d caused this inconsistency. Instead of "climatological field" as originally described, it should be "(d) 500-hPa horizontal temperature advection computed by anomalous wind field with climatological temperature gradient (shading) and climatological wind field with anomalous temperature gradient (contours, unit: K/day)". The shading in Fig. 4d, which represents the 500-hPa warm advection anomalies driven by the anomalous wind field, closely resembles the corresponding meridional wind anomalies (Fig. 4e, color shading). Therefore, our claim remains unchanged.

We have corrected this error in the caption of Fig. 4. We are sorry for this mistake. The manuscript has been thoroughly re-checked. We thank the reviewer for pointing this out.

Moreover, the precipitation anomaly distributes along a zonal-oriented band rather than along a meridional-oriented band, which suggests that the warm advection by the zonal wind is likely stronger than that by meridional wind. I agree with the moisture transport role of the southerly wind, but doubt the dominant role of the southerly wind in the warm advection at 500 hPa over

East Asia. I suggest that the authors present the spatial pattern and quantitative result of temperature advection by different components.

Response: Thanks for the comments. Our reply is arranged as below.

i. Spatial pattern of temperature advection

To address the question regarding the dominant influence on warm advection, specifically concerning the zonal orientation of the rain band, Fig. R3–1 provides more information. It shows the 500-hPa spatial pattern of total temperature advection anomalies, as well as the associated anomalous wind-driven temperature advection for each zonal and meridional component.

The spatial pattern of meridional wind-driven temperature advection anomalies plays a prominent role in the total temperature advection anomalies (Fig. R3–1), as this component is shown to spatially overlap in the total temperature advection anomalies during the lag days. In contrast, the zonal wind-driven temperature advection anomalies are relatively weak and not well-organized, especially in the lag days. This is consistent with our main claim that the meridional wind is important in connecting the ISM and the EASM (i.e., the CGT wave train), steering the warm advection and upward motion towards East Asia (Figs. 2 and 4, and the corresponding description in line 199–203 in original manuscript or also line 199–203 in the revised manuscript).

Fig R3–1. Evolution of HIST lead–lag composite 500-hPa temperature advection anomalies for each cluster (columns from left to right, respectively). The color shading represents the total 500-hPa temperature advection anomalies, and the black and blue contours indicate the anomalous meridional and zonal wind-driven 500-hPa temperature advection, respectively. Dotted regions denote anomalies significant at the 99% confidence level based on the two-sided Student’s t test.

ii. Quantitative result of temperature advection by different components

The quantitative result of the temperature advection anomalies by different components is shown in Fig. R3–2. The results confirm the dominant role of the anomalous meridional wind-driven warm advection in the total warm advection anomalies, especially in the lag days of each cluster, in response to the CGT wave train in the East Asian sector induced by the ISM heating. The cold advection, on the other hand, is mainly contributed by the anomalous temperature field in the lead days, which is associated with negative geopotential heights near the wet band in clusters 2 and 3. These results are again consistent with Figs. 4c–d and the corresponding description in line 199–203 of the original manuscript (line 199–203 in the revised manuscript).

Fig R3–2. Temporal evolution of anomalies for different 500-hPa temperature advection components from lead10 to lag10, averaged over the region of 30°N, 105°–140°E for clusters 1–4 (columns from left to right, respectively). The upper panels present the meridional components of 500-hPa temperature advection driven by anomalous meridional wind with climatological temperature gradient (dark gray bars) and anomalous temperature gradient with climatological meridional wind (light gray bars). Similarly, the bottom row shows the zonal components. The black curves represent the total 500-hPa temperature advection anomalies. The bars with black borders and thick black line denote anomalies significant at the 99% confidence level based on the two-sided Student's t test.

The Fig. R3–1 is added to the supporting information as supplementary Fig. 8, together with related description added in line 204–205.

2. This study has shown the effect of ISM heating on early-summer precipitation in East Asia, on the sub-seasonal timescales. The results are important for understanding the sub-seasonal variability of the East Asian summer precipitation. I wonder if the authors can quantify the effect of the ISM heating. I am curious about the two aspects: (1) probability for wet events in the north wet band due to the ISM heating and (2) ratio of intraseasonal precipitation in the north wet band can be explained by the ISM heating. The quantitative results may provide a straightforward view of the ISM heating on sub-seasonal precipitation variability in East Asia.

Response Thank you for your insightful comments. The results, as illustrated in Fig. R3–3, present quantitative insights. Figure R3–3 compares the probability distribution of daily wet band precipitation anomalies created using the entire study period (i.e., all days from June and first 2 pentad of July) and that using only lag0–lag10 days as the period contains the ISM influence. In this figure, positive wet band precipitation anomalies greater than 0.5 mm/day is regarded as wet band events. Based on this figure, our interpretations about the two suggested aspects are described as below:

i. Probability for wet events in the north wet band due to the ISM heating

Among all wet band events, the probability of the wet band events from June to early July with presence of lag0–lag10 days (i.e., influenced by the strong June ISM heating) is 21.02% for observations and 16.72% for HIST. Noteworthy, these probabilities rise to 23.26% and 19.77%, and further to 31.58% and 24.67% for the events exceeding +1 and +2 standard deviations, respectively, of wet band precipitation anomalies. Therefore, the probability of wet band events coincident with abnormal ISM heating increases with increasing precipitation anomaly (Figure R3–3). This finding suggests that the ISM plays a more prominent role in stronger wet band precipitation anomalies.

ii. Ratio of subseasonal precipitation in the north wet band can be explained by the ISM heating

As in Fig. R3–3, it is clear that the wet band precipitation anomalies during lag0–lag10 days tend to shift to positive values (Figure R3–3). The mean wet band precipitation anomalies are 1.04 mm/day for observations and 0.66 mm/day for HIST, meaning the magnitude of wet band precipitation anomalies during lag0–lag10 days is greater than those of climatology.

Regarding the exceedance probabilities of +1 standard deviation of wet band precipitation anomalies during lag0–lag10 days are 1.83 times higher in observations and 1.59 times higher in HIST, respectively, in comparison to the probability estimated for daily wet band precipitation anomalies in June and early July. Therefore, it indicates the likelihood of wet band event is higher when abnormal ISM heating exists.

We have added Fig. R3–3 as Fig. 6 in the revised manuscript, and associated descriptions in 266–288.

Fig R3–3. The probability distribution of wet band precipitation anomalies during lag0–lag10 days relative to all days of the entire study period (i.e., days from June and first 2 pentads of July), based on (a) observations and (b) HIST. The bars denote the count of occurrences (left Y-axis) for every 1 mm/day bin, and percentage ratio of wet band occurrences coinciding with lag days is indicated at the top of each bar. The value of +1 standard deviation of June to early July wet band precipitation anomalies is illustrated by grey dashed lines. The curves represent the probability density (right Y-axis), with the mean values at the top right of each panel. The occurrence of the wet band is defined as positive precipitation anomaly greater than 0.5 mm/day. The domain for the wet band is depicted as the green rectangle in Supplementary Fig. 1.

3. One major conclusion of this study is that the jet structures modulate the role of ISM heating as a trigger or reinforcer of the rainband. The authors do show different preceding signals in the mid-high latitude Eurasia for four different jet structures (Fig. 5). Though the signals are clear, I wonder the underlying mechanisms. I understand the different locations of jet axis and different jet intensity may affect the latitudinal and longitudinal locations of the CGT portion over East Asia. But how can the jet structures in clusters 1 and 4 lead to preexisting southerly winds and warm advection, but that in clusters 2 and 3 cannot?

Response: Thank you for your question. As we stated in Line 161–164 in the original manuscript, we have observed that the East Asian jets prior to the extreme June ISM heating (i.e., event0) exhibit characteristic of jet meridional displacement (JMD) depending on the cluster. The JMD is known as the leading pattern for summer East Asian jet (e.g., Hong and Lu, 2016; Chowdary et al., 2021).

In clusters 1 and 4, the jet runs straight towards East Asian sector (Fig. 3 in the original

manuscript). In these clusters, the vertical structure of zonal wind anomalies displays a typical northward JMD, with corresponding north-positive and south-negative zonal wind anomalies extending from the upper troposphere to near surface (Figs. R3–4a and d). The northward JMD accompanies the strong southerly wind (e.g., Wang et al., 2018), as we also observed in clusters 1 and 4 whose southerly winds prevail in the troposphere (Figs. R3–4a and d). It also shows the region with strong southerly winds coincides with anomalous temperature advection (Figs. R3–4a and d). In contrast, clusters 2 and 3 exhibit a southward bending jet flow (Fig. 3 in the original manuscript), and we found reversed tropospheric zonal wind anomalies compared to clusters 1 and 4, indicating a southward JMD feature (Figs. R3–4b and c). In these clusters, the meridional wind and temperature advection anomalies in the troposphere are not clearly seen near the wet band area.

In conclusion, the major difference about the East Asian jet pattern over prior to the extreme ISM heating between clusters 1 and 4 and clusters 2 and 3 is the jet meridional displacement. The jet displacement to north as in clusters 1 and 4 leads to warm advection sustained by southerly wind.

Furthermore, we found that daily wet band precipitation in June is evidently correlated with both the 500-hPa meridional wind and the northward JMD (Figs. R3–5a, b and c). The subseasonal lead–lag correlations between them, however, indicate that the southerly wind occurred prior than wet band precipitation, and the JMD is likely to occur later than the appearance of both the southerly wind and wet band (Figs. R3–5d). Therefore, the jet structure with northward JMD is possibly a sign for the existed southerly wind and wet band, which lead to the upcoming ISM forcing as a reinforcer, and vice versa as a trigger for clusters 2 and 3.

The Fig. R3–4 and Fig. R3–5 is added to supplementary as Fig. 5 and Fig. 9, and related descriptions are added in line 160–165 and 212–216.

Fig R3-4. Cross section along 120°E showing the anomalous wind field of 4 clusters on event0 days. The color shading represents zonal wind anomalies, the arrows indicate meridional wind with vertical velocity, and the black and red contours depict reconstructed wind patterns and temperature advection anomalies, respectively. Stippled regions and arrows indicate anomalies that are statistically significant at the 99% confidence level.

Fig R3-5. The spatial pattern of HIST June standardized daily area-mean wet band precipitation (Npr) regression against (a) precipitation, and 500-hPa (b) meridional wind (V500), and (c) zonal wind (U500). (d) June subseasonal lead-lag correlations among Npr, area-averaged 500-hPa meridional and zonal winds near the wet band. The V500 lead-lag Npr means lead-lag correlations of 500-hPa meridional wind with respect to Npr, similarly for others. Values with solid circle satisfy $p < 0.01$. The green rectangle denotes the wet band area, similar to Supplementary Fig. 1.

Minor comments:

1. Line 24: Why is the northward tilt of southerly wind over East Asia so important to be presented in the abstract? In addition, please explain the possible reason.

Response: We acknowledge the reviewer for this comment. We realize that this aspect is not directly related to our major findings. We have removed it from the abstract.

2. Fig. 5: U200 anomalies are absent.

Response: Corrected.

3. The authors used two composite analysis based on event0 and lag0 day, in which the former is only used for the jet cluster analysis. Why were the two different composite methods used? The different composite result cannot be explained directly for each other. If possible, please use one for all the analysis in one paper. Otherwise, the reason and the possible effect using different composites on results should be discussed.

Response: Thanks for the comment. As pointed out by the reviewer, the lag0 day (i.e., maximum ISM heating) is used in all lead-lag composites to study how the EASM wet band responds to ISM heating. In this study, we also aim to understand the interaction between ISM heating and the jet stream as it is commonly accepted that the jet stream acts as a waveguide that possibly modulate the connection between ISM and EASM. To discuss the response of the wave propagation initiated by atmospheric heating over ISM regions, we desired to characterize the jet pattern in prior to the emergence of ISM heating. Hence, event0 was defined as the time when the ISM heating starts to increase in each event. This reason and the concept were described in line 149–151 and line 402 in the original manuscript, and for further clarification, more descriptions were added to line 435–437 in the revised manuscript.

We here conducted an additional analysis to show the relationship between event0 day and lead-lag day. The mean duration between event0 and lag0 is about 4 days (Fig. R3–6; $C1 = 4.16$ days, $C2 = 4.10$ days, $C3 = 3.83$ days, and $C4 = 4.36$ days). Therefore, event0 roughly corresponds to lead4 day (i.e., 4 days before lag0). To confirm the similarity between event0 and lead4, cluster labels are assigned for each lead4 sample considering the minimum Euclidean distances relative to the four reference patterns as shown in Fig. 3. Then, the lead–lag composite is conducted for newly assigned cluster labels based on lead4. The results (Fig. R3–7) are similar to the composite created for event0 (Fig. 4 in the original manuscript). Thus, lead4 could be regarded as event0, which is, in other words, the jet classification in this study was performed for the jet pattern of approximately 4 days prior to lag0. We believe this is a reasonable setting that meets with our original motivation which is to monitor how preconditioning jet structure affects the connection between ISM and EASM in the following days. We added this discussion in the revised manuscript (line 451–454), and Figs. R3–6 is added to the supporting information as supplementary Fig. 16 which would be helpful for readers to recognize how these two composites are related.

Fig R3–6. Probability distribution of the number of days between event0 and lag0 in each cluster.

Fig R3–7. Similar to Fig. 4 in the original manuscript but based on the new clusters from samples of lead4 days.

Reference:

- Chowdary, J. S., Vibhute, A. S., Darshana, P., Parekh, A., Gnanaseelan, C., & Attada, R. (2021). Meridional displacement of the Asian jet and its impact on Indian summer monsoon rainfall in observations and CFSv2 hindcast. *Climate Dynamics*, 58(3–4), 811–829. <https://doi.org/10.1007/s00382-021-05935-1>
- Hong, X., & Lu, R. (2016). The meridional displacement of the summer Asian jet, Silk Road pattern, and tropical SST anomalies. *Journal of Climate*, 29(10), 3753–3766. <https://doi.org/10.1175/JCLI-D-15-0541.1>
- Wang, S., Zuo, H., Zhao, S., Zhang, J., & Lu, S. (2018). How East Asian westerly jet's meridional position affects the summer rainfall in Yangtze-Huaihe River Valley? *Climate Dynamics*, 51(11–12), 4109–4121. <https://doi.org/10.1007/s00382-017-3591-3>

REVIEWERS' COMMENTS

Reviewer #3 (Remarks to the Author):

The authors have addressed all my concerns well. I have only one minor comment: What is the meaning and purpose of the numbers in percentage (19.47%,...) on the grey bars in Fig. R3-3?

Reviewer #3 (Remarks to the Author):

The authors have addressed all my concerns well.

Response: We sincerely appreciate the time and effort the reviewer has taken to thoroughly read and consider our manuscript. We are grateful to the reviewer for bringing constructive questions and comments to our attention.

I have only one minor comment: What is the meaning and purpose of the numbers in percentage (19.47%, ...) on the grey bars in Fig. R3-3?

Response: We have revised the captions to make them clearer. They represent the percentage ratio of wet band occurrences of the entire study period coinciding with lag days at each 1 mm/day precipitation anomalies interval. Therefore, we conclude that the probability of wet band events coincident with abnormal ISM heating increases with increasing precipitation anomalies (Figure R3–3). This finding suggests that the ISM plays a more prominent role in stronger wet band precipitation anomalies (line 283 in the manuscript). Here we think these values will be helpful for quantitative understanding.

Fig R3–3. The probability distribution of wet band precipitation anomalies based on (a) observation and (b) large ensemble historical climate experiment (HIST). Grey and green colors are for all days of the entire study period (i.e., days from June and first 2 pentads in July) and for days during lags of 0–10 days, respectively. The bars denote the count of occurrences (left Y-axis) for every 1 mm/day bin, and percentage value at the top of each bar refers to wet band occurrences coinciding with lags of 0–10 days. The occurrence of the wet band is defined as positive precipitation anomaly greater than 0.5 mm/day. The domain for the wet band is depicted as the green rectangle in Supplementary Fig. 1. The value of +1 standard deviation of June to early July wet band precipitation anomalies is illustrated by dashed grey vertical line. The curves represent the probability density (right Y-axis), with the mean values at the top right of each panel.